# Exploiting Presentative Feature Distributions for Parameter-Efficient Continual Learning of Large Language Models

Xin Cheng [1] [*]   Jiabo Ye [2]   Haiyang Xu [2]   Ming Yan [2]   Ji Zhang [2]   Feng Liu [3]   Fei Huang [2]   Lei Feng [1]

## Abstract

Endowing large language models (LLMs) with continual learning (CL) capacities is practically important, which enables them to dynamically acquire new knowledge over time. Although many effective methods have been proposed for CL of LLMs, they did not consider online scenarios, thereby sharing a common problem: *information leakage* (IL), where the task-related information of learned tasks is accessed or reused again. IL not only imposes potential risks on data privacy protection but also significantly hinders the deployment of LLMs in real-world scenarios. To avoid IL while maintaining outstanding CL performance, we propose a novel CL method for LLMs, which first characterizes a parameter-efficient fine-tuning (PEFT) block by a presentative feature distribution, and then dynamically selects the appropriate PEFT blocks for each instance based on its similarity with the presentative feature distributions. Extensive experiments validate the effectiveness of our method on the CL of LLM, showcasing its potential to enhance both privacy and adaptability in practical applications. Our source code is available at https://github.com/ZERO-9215/Online-CL-LLMs.

## 1. Introduction

In recent years, large language models (LLMs) (Brown et al., 2020; Raffel et al., 2020; Touvron et al., 2023; Achiam et al., 2023) have made remarkable progress in their ability to address a variety of problems. At the same time, LLMs need to be updated regularly to accurately reflect the evolving knowledge, needs, and values of humanity (Biesialska et al.,

2020; Ouyang et al., 2022). However, updates to LLMs typically involve adding new data and retraining the model, which is highly inefficient and inflexible. Therefore, endowing LLMs with continual learning (CL) (Lopez-Paz & Ranzato, 2017; Chaudhry et al., 2019; Wu et al., 2021; Ke & Liu, 2022; Wang et al., 2023a; Chen et al., 2023; Zhao et al., 2024; Wu et al., 2024) capabilities is crucial for their deployment in real-world scenarios, which enables them to dynamically adapt to new tasks and acquire additional knowledge over time (Luo et al., 2023; Zhai et al., 2023).

The vast number of parameters in LLMs imposes a significant computational burden on CL (Wu et al., 2024), rendering traditional CL methods unsuitable for direct application. To avoid exorbitant training overhead, recent research has begun exploring CL for LLMs using parameter-efficient fine-tuning (PEFT) (Houlsby et al., 2019; Lester et al., 2021; Hu et al., 2021). A mainstream approach is parameter isolation-based methods (Rusu et al., 2016; Fernando et al., 2017; Zhao et al., 2024). These methods allocate a new PEFT block for each new task to capture task-specific knowledge, which ensures that the parameters learned by each task do not interfere with each other and are reorganized later. Although many effective CL methods for LLMs using PEFT have been proposed, they are not designed for online learning scenarios where tasks arrive in a stream, meaning that accessing or reusing task-related information (e.g., training data and task identifiers) from previously learned tasks is not allowed. Consequently, previous methods share a common problem: *Information Leakage* (IL) from previously learned tasks to downstream tasks.

Many CL methods (McCloskey & Cohen, 1989; Kirkpatrick et al., 2017; Kotha et al., 2024) simplify the CL setting to avoid forgetting by assuming that the task identity of test instances (i.e., which task they come from) is known during testing. However, in the real world, we typically have access only to model weights, and often it is unclear which tasks have been learned, let alone which tasks the test instances come from. Some other methods (Lopez-Paz & Ranzato, 2017; Shin et al., 2017; Isele & Cosgun, 2018; Yin et al., 2022; Scialom et al., 2022; Mok et al., 2023; Feng et al., 2024) to mitigate catastrophic forgetting assume that the model can access and reuse the training data of learned

---

[*]Work done during internship at Alibaba Group. [1]School of Computer Science and Engineering, Southeast University, China [2]Alibaba Group [3]TMLR Group, University of Melbourne. Correspondence to: Lei Feng <fenglei@seu.edu.cn>.

*Proceedings of the 42$^{nd}$ International Conference on Machine Learning*, Vancouver, Canada. PMLR 267, 2025. Copyright 2025 by the author(s).

| Avoid IL | SuperNI Benchmark | | Long Sequence Benchmark | |
|---|---|---|---|---|
| | SAPT | TaSL | SAPT | TaSL |
| ✘ | 51.54 | 53.01 | 82.02 | 84.33 |
| ✔ | 11.12 | 27.43 | 10.18 | 72.29 |
| Gaps | -40.42 | -25.58 | -71.84 | -12.04 |

Table 1: The performance of the two best-performing CL methods (SAPT and TaSL) using the T5-Large model on two CL benchmark (SuperNI Benchmark and Long Sequence Benchmark). ✘ indicates methods with IL, while ✔ signifies methods without IL.

tasks, for example, by regularly replaying this data during training. These assumptions not only require overhead for storing data from learned tasks, but also necessitate periodic computational costs for replay, which pose scalability challenges for CL. Furthermore, these assumptions also hinder the application of CL in scenarios involving data-sensitive or specialized tasks, where accessing data-related information is not allowed. In table 1, we show that even for the current state-of-the-art CL methods, there is a performance gap ranging from about 12.04% to 71.84% between with and without IL. Therefore, developing more effective online CL methods for LLMs avoiding IL is crucial for the safe and effective application and deployment of CL for LLMs.

To address the problem of IL, we propose a novel online method exploiting presentative feature distributions for parameter-efficient CL of LLMs. Our method first leverages the feature representation capability of well-developed pre-trained LLMs trained on large-scale data to encode data domain information of tasks into a presentative feature distribution. This presentative feature distribution is used to characterize learned PEFT blocks relevant to the task. Then, we calculate the similarity between instances and the stored presentative feature distributions, using this similarity to dynamically select the associated PEFT blocks. During training, these blocks are treated as prior knowledge to learn new tasks, while during testing, the selected PEFT blocks will be combined to predict. At all times, our method only accesses the high-dimensional and human-incomprehensible presentative feature distributions statistically derived from the pre-trained LLMs, hence it ensures that no information is leaked. Moreover, because the presentative feature distribution is only associated with the pre-trained LLMs and the related PEFT block, our method demonstrates strong expandability. It allows learned PEFT blocks and presentative feature distributions on different tasks using the same model framework to be directly expanded without additional training. Without IL, our method achieves state-of-the-art performance, approaching or even surpassing the performance of methods with IL.

Our contributions to CL can be summarized as follows:

- We empirically found significant performance differences between previous CL methods with and without IL and identified the cause of this difference as the introduction of a new risk of forgetting during the selection process.

- We propose a simple and effective online CL method for LLMs, which utilises the feature representation capabilities of pre-trained LLMs to characterize each PEFT block by encoding a presentative feature distribution.

- Based on our proposed method, we provide an extension that is able to combine PEFT blocks and presentative feature distributions learned by our method using the same model architecture without additional training.

Extensive experiments validate that our method avoids IL while maintaining outstanding CL performance.

## 2. Background

### 2.1. Continual Learning

Let $\mathcal{T} = \{(\boldsymbol{x}_i, \boldsymbol{y}_i)\}_{i=1}^n$ be the target task with the size of $n$, and each example $(\boldsymbol{x}_i, \boldsymbol{y}_i) \in \mathcal{X} \times \mathcal{Y}$ is assumed to be sampled from an unknown data distribution with probability density $p(\boldsymbol{x}, \boldsymbol{y})$. The goal of continual learning (CL) (Wu et al., 2024) is to train a single model $f$ to adapt a sequence of tasks $\{\mathcal{T}_1, \mathcal{T}_2, \dots \mathcal{T}_K\}$ that arrive in a streaming fashion, where model $f$ can only access the $k$-th task $\mathcal{T}_k = \{(\boldsymbol{x}_i^k, \boldsymbol{y}_i^k)\}_{i=1}^{n_k}$ in the $k$-th time step. Then, the optimal objective of CL is the following:

$$\max_f \sum\nolimits_{k=1}^K \mathbb{E}_{p(\boldsymbol{x}^k, \boldsymbol{y}^k)} \log p(\boldsymbol{y}^k | \boldsymbol{x}^k, f). \quad (1)$$

Previous CL research posits that an effective CL method should address two primary challenges: 1) Catastrophic Forgetting (CF) (McCloskey & Cohen, 1989; Kirkpatrick et al., 2017; Kotha et al., 2024), where the performance on learned tasks significantly deteriorates when learning new tasks; and 2) Knowledge Transfer (KT), which includes both forward transfer and backward transfer. Forward transfer refers to the ability to leverage knowledge from learned tasks to achieve better performance on a new task. Backward transfer refers to the ability of learned tasks to benefit from the knowledge acquired while learning new tasks, thereby improving their performance.

### 2.2. Parameter-Efficient Fine-Tuning

Pre-trained LLMs developed using large-scale pre-training datasets have become a powerful foundation for addressing a variety of target tasks. When faced with a specific downstream task, pre-trained LLMs can be adapted to effectively solve the task through parameter-efficient fine-tuning (PEFT) (Houlsby et al., 2019; Lester et al., 2021; Li & Liang,

2021; He et al., 2021; Ding et al., 2022; Zaken et al., 2022). Low-Rank Adaptation (LoRA) (Hu et al., 2021) is the most popular PEFT method. It assumes that when the pre-trained model is adapted to the new target task, the changes in model parameters reside within a low-rank space. Consequently, we only need to update the low-rank weights associated with the parameters during adaptation, eliminating the need to update all parameters. Specifically, for a pre-trained weight matrix $\mathbf{W} \in \mathbb{R}^{out \times in}$ and an input feature $h(\boldsymbol{x}) \in \mathbb{R}^{in}$, LoRA adds a low-rank decomposition block $\mathbf{BA}$ for output $\hat{\mathbf{W}} h(\boldsymbol{x})$ as follows:

$$\hat{\mathbf{W}} h(\boldsymbol{x}) = \mathbf{W} h(\boldsymbol{x}) + \mathbf{BA} h(\boldsymbol{x}), \qquad (2)$$

where $\mathbf{A} \in \mathbb{R}^{d \times in}$, $\mathbf{B} \in \mathbb{R}^{out \times d}$, and the rank $d \ll \min(in, out)$. During weight updates, the pre-trained weight matrix $\mathbf{W}$ remains fixed, and only the LoRA block $\mathbf{A}$ and $\mathbf{B}$ are updated.

### 2.3. Continual Learning for LLMs with PEFT

Large language models (LLMs) with billions of parameters impose a significant computational burden on CL (Wu et al., 2024). To avoid exorbitant training overhead, recent research has begun exploring CL for LLMs using PEFT.

A key technique is knowledge isolation-based method, which requires the model to learn new knowledge without modifying what has already been learned. Orthogonal Low-Rank Adaptation (O-LoRA) (Wang et al., 2023a) enforces that the LoRA parameters learned for new tasks are orthogonal to the existing LoRA parameters. TaSL (Feng et al., 2024) decomposes LoRA blocks into $d$ sub-blocks and selectively updates these sub-blocks when learning new tasks to avoid conflicts with learned tasks.

Another important technique is parameter isolation-based methods (Rusu et al., 2016; Fernando et al., 2017), which allocate a new PEFT block for each new task. SAPT (Zhao et al., 2024) designed a shared attention framework to dynamically learn and select LoRA blocks. However, this framework introduces new shared parameters (Query Projection), which still poses a risk of forgetting. To address this issue, Zhao et al. (2024) further proposed a data replay method using pseudo-samples.

## 3. Proposed Method

In this section, we first reflect on why previous CL methods show performance differences with and without IL. We then propose a novel and expandable method that avoids IL while maintaining outstanding CL performance.

**Notations.** In this paper, we use LoRA fine-tuning as our main PEFT implementation. For a given pre-trained model $f$ and a target task $\mathcal{T}_k = \{(\boldsymbol{x}_i^k, \boldsymbol{y}_i^k)\}_{i=1}^{n_k}$, we denote

$\mathbf{W}^l \in \mathbb{R}^{out \times in}$ as the weight matrix for layer $l$ of model $f$ and use $\mathbf{A}_k^l \in \mathbb{R}^{d \times in}$ and $\mathbf{B}_k^l \in \mathbb{R}^{out \times d}$ to represent the LoRA blocks learned for task $\mathcal{T}_k$ in layer $l$.

### 3.1. Revisiting the Causes of Performance Differences

For parameter-based isolation CL methods, the CL process can be divided into two stages: 1) Selection: choosing relevant PEFT blocks for the instance, and 2) Merging: combining the selected PEFT blocks. The state-of-the-art parameter-based isolation method, SAPT (Zhao et al., 2024), designs a shared attention framework to learn how to select and merge LoRA blocks. However, this shared attention framework introduces new trainable parameters, known as query projections, which may also introduce a new forgetting problem. In Figures 1.(a) and 1.(b), we show the heatmap of SAPT selecting LoRA blocks with and without IL using query projection. We can observe that without IL, SAPT extremely selects the LoRA of the last task for all task instances, indicating the occurrence of forgetting, which validates our hypothesis. Although SAPT uses a reflection module to replay the pseudo-samples in order to mitigate the new forgetting, this can also lead to IL. To solve the problem, we should avoid introducing new trainable parameters in the selection stage.

Inspired by the technique of fine-tuning (Howard & Ruder, 2018; Kumar et al., 2022; Parthasarathy et al., 2024), zero-shot (Radford et al., 2021; Kojima et al., 2022), in-context learning (Dong et al., 2022) and pre-training model initialization (Touvron et al., 2023; Dubey et al., 2024), where well-developed pre-trained LLMs are repurposed, we can similarly leverage pre-trained LLMs for selection without introducing new trainable parameters. Following this idea, we propose a novel online CL method, which leverages the feature representation capabilities of pre-trained LLMs to characterize each PEFT block by encoding a presentative feature distribution. This method mainly consists of three key components: Feature Distribution Module, Similarity Module, and Dynamic Selection Module. The overall architecture of our proposed method is illustrated in Figure 2.

### 3.2. Feature Distribution Module

We assume that pre-trained LLMs trained on a large-scale general dataset already possess robust feature representation capability, which has been demonstrated in zero-shot, few-shot, and in-context learning. Therefore, the feature space of the pre-trained LLMs itself can be utilized to distinguish and categorize different tasks and knowledge, such that we do not need to train a new classifier. Specifically, we denote by $D_k^l$ the presentative feature distribution of task $\mathcal{T}_k$ at layer $l$, and $D_k^l$ is computed as follows:

$$D_k^l = \mathbb{E}_{p(\boldsymbol{x}^k, \boldsymbol{y}^k)}[\mathbf{W}^l h^l(\boldsymbol{x}^k)], \qquad (3)$$

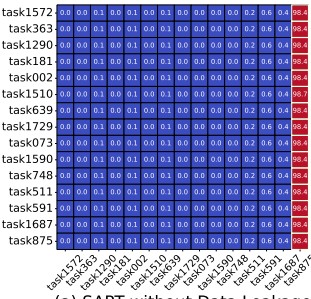
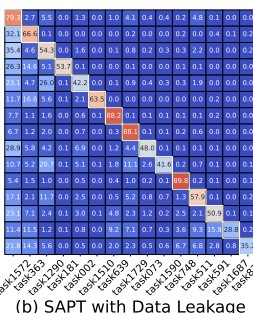
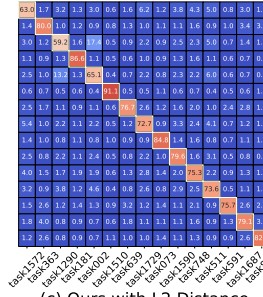
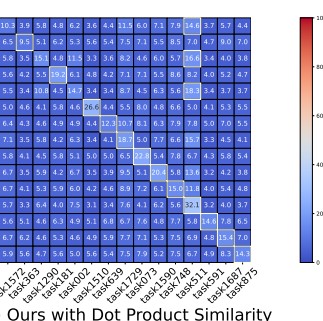

|  (a) SAPT without Data Leakage | (b) SAPT with Data Leakage | (c) Ours with L2 Distance | (d) Ours with Dot Product Similarity |

Figure 1: The heatmap demonstrates the selection of LoRA on the SuperNI Benchmark across 15 tasks. Figures (a) and (b) illustrate the results of the state-of-the-art parameter-based isolation CL method, SAPT, with and without IL, utilizing the query projection for selecting LoRA. Figures (c) and (d) display the results of our proposed method using L2 distance and dot product similarity for selecting LoRA (averaged across all model layers). For all results, we scaled the values by a factor of 100 and highlighted the highest value in each row with a yellow box.

where $h^l(\boldsymbol{x}^k)$ denotes the hidden state input of instance $\boldsymbol{x}^k$ in layer $l$. The presentative feature distribution $D_k^l$ primarily depends on the weight matrix $\mathbf{W}^l$ of the pre-trained LLMs and the data distribution $p(\boldsymbol{x}^k, \boldsymbol{y}^k)$. Since the weight matrix $\mathbf{W}^l$ is frozen during LoRA fine-tuning, the presentative feature distribution $D_k^l$ varies only with the data distribution, thereby being regarded as a mapping of the data distribution by the pre-trained LLMs. Additionally, based on parameter-based isolation techniques, we allocate a new LoRA block for each streaming arrival task. Consequently, the presentative feature distribution can also be used to characterize the corresponding LoRA block.

### 3.3. Similarity Module

For a reliable presentative feature distribution $D_k^l$ and an arbitrary instance $\boldsymbol{x}$, if $\boldsymbol{x}$ is related (e.g., instruction or knowledge) to task $\mathcal{T}_k$, then the feature of $\boldsymbol{x}$ at layer $l$ (i.e., $\mathbf{W}^l h^l(\boldsymbol{x}^k)$) will be close to $D_k^l$, that is, with greater similarity. Building on this idea, we denote the similarity between instance $\boldsymbol{x}$ and presentative feature distribution $D_k^l$ as $\Phi(\boldsymbol{x}, D_k^l)$. In this paper, we explore the following two classic methods for computing the similarity:

- Negative $L_2$ Euclidean Distance:

$$\Phi_{L_2}(\boldsymbol{x}, D_k^l) = -\sqrt{||\mathbf{W}^l h^l(\boldsymbol{x}) - D_k^l||^2}; \quad (4)$$

- Dot Product Similarity:

$$\Phi_{\text{Dot}}(\boldsymbol{x}, D_k^l) = \frac{\mathbf{W}^l h^l(\boldsymbol{x}) \cdot D_k^l}{\sqrt{\text{out}^l}}, \quad (5)$$

where $\text{out}^l$ donates the output dimension of layer $l$.

### 3.4. Dynamic Selection Module

Using the feature distribution and similarity module, we can easily obtain the dynamically selected and merged out-

put for instance $\boldsymbol{x}$ of $k$-th task in layer $l$ by normalizing the similarities of instance $\boldsymbol{x}$ with existing LoRA blocks $\{\mathbf{B}_1^l \mathbf{A}_1^l, \mathbf{B}_2^l \mathbf{A}_2^l \ldots \mathbf{B}_k^l \mathbf{A}_k^l\}$. The output $\hat{\mathbf{W}}^l h^l(\boldsymbol{x})$ of instance $\boldsymbol{x}$ at layer $l$ is given as follows:

$$\mathbf{W}^l h^l(\boldsymbol{x}) + \sum_{j=1}^{k} \frac{\exp(\Phi(\boldsymbol{x}, D_j^l)/T)}{\text{Sum}(\exp(\Phi/T))} \mathbf{B}_j^l \mathbf{A}_j^l h^l(\boldsymbol{x}), \quad (6)$$

where $\text{Sum}(\exp(\Phi/T)) = \sum_{v=1}^{k} \exp(\Phi(\boldsymbol{x}, D_v^l)/T)$ and $T$ is the temperature coefficient used to control normalization. During the training for the $k$-th task $\mathcal{T}_k$, only the LoRA blocks $\mathbf{A}_k^l$, $\mathbf{B}_k^l$ and presentative feature distribution $D_k^l$ associated with the current task $\mathcal{T}_k$ are updated, while the other parameters (learned task-specific LoRA blocks $\mathbf{A}_j^l$, $\mathbf{B}_j^l$ and presentative feature distributions $D_j^l$, and the pre-trained model matrix $\mathbf{W}^l$) remain frozen. It is worth noting that the update of the presentative feature distribution $D_k^l$ is accomplished statistically as described in Eq. 3, without introducing any additional trainable parameters, thus not introducing a new forgetting problem. Additionally, the dynamic selection module can selectively activate some LoRA blocks rather than activating all of them, as shown in the Top-$K$ selection method below:

$$\sum_{j \in \text{Top-}K(\Phi(\boldsymbol{x}, D^l))} \frac{\exp(\Phi(\boldsymbol{x}, D_j^l)/T)}{\text{Sum}(\exp(\Phi/T), \text{Top-}K)} \mathbf{B}_j^l \mathbf{A}_j^l h^l(\boldsymbol{x}),$$

where $\text{Sum}(\exp(\Phi/T), \text{Top-}K) = \sum_{v \in \text{Top-}K(\Phi(\boldsymbol{x}, D^l))} \exp (\Phi(\boldsymbol{x}, D_v^l)/T)$. Selective activation can filter out some irrelevant LoRA blocks and focus on the most relevant ones. We will discuss the selection of Top-$K$ in the following experiments section.

Through feature distribution, similarity, and dynamic selection modules, our method achieves online CL for LLMs, effectively avoiding IL while addressing two key challenges

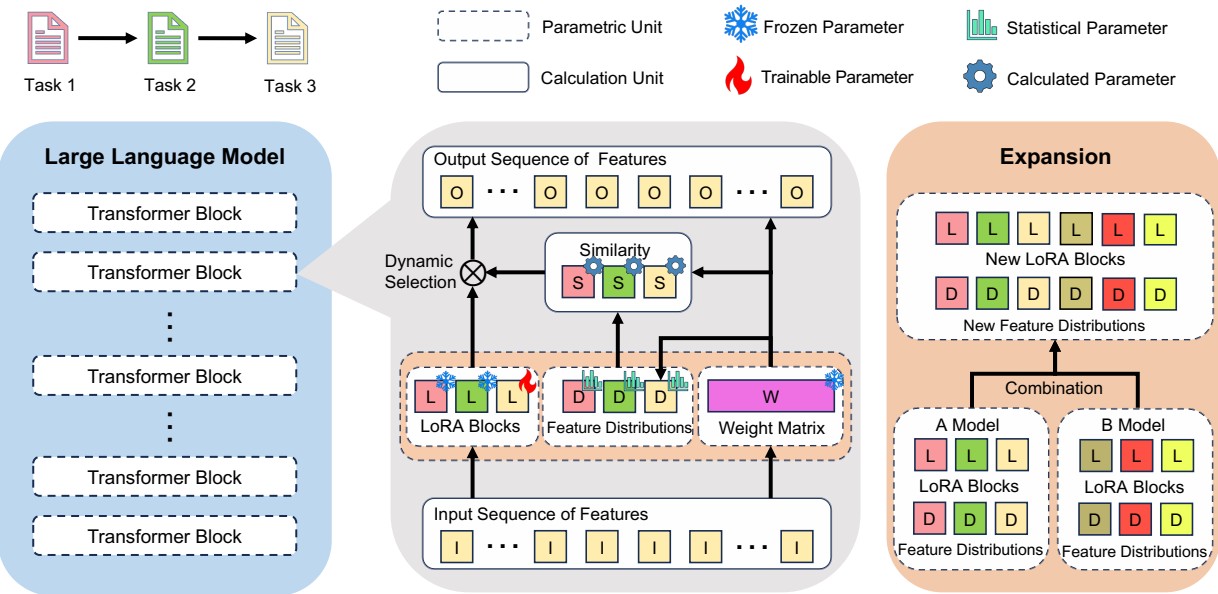

Figure 2: The overall architecture of our proposed method. We assume that tasks 1 and 2 have already been learned, and we are currently learning task 3. The central part of the figure illustrates the continual learning process of our proposed method: 1) The features of the input instances first pass through the frozen weight matrix of the pre-trained LLMs to obtain hidden states, which are then used to update the presentative feature distribution relevant to Task 3; 2) The resulting hidden states are compared with existing feature distributions to calculate similarity; 3) Based on the calculated similarity, LoRA blocks are dynamically selected and output is obtained. The rightmost part of the figure shows the expansion of LoRA blocks and feature distributions across different trained models with the same architecture.

that affect CL performance. 1) IL: During both training and testing, our method only accesses the presentative feature distributions and LoRA blocks, completely avoiding access to any data information from learned tasks, including pseudo or synthesized data. Furthermore, the presentative feature distributions are high-dimensional and incomprehensible to humans. Moreover, presentative feature distributions represent averaged information across populations and cannot reconstruct individual information, ensuring that no related information is exposed. 2) CL: our method is a parameter isolation-based method. It ensures that task-relevant knowledge is fully retained while dynamically selecting appropriate knowledge to prevent forgetting. Additionally, our method does not introduce any new trainable parameters, as the presentative feature distribution is derived statistically, thereby avoiding new forgetting issues. 3) KT: For forward transfer, when learning a new task, our method can effectively learn the new task by selecting relevant learned task knowledge as prior knowledge through the dynamic selection module. During testing, our method compares the features of instances with the presentative feature distributions to extract knowledge relevant to the test instance. This enables learned tasks to benefit from the knowledge gained in new tasks, achieving backward transfer.

## 3.5. Flexible Expansion Advantage

Our proposed method characterizes each LoRA block with the presentative feature distribution based on the robust feature representation capability of the well-developed pre-trained LLMs, retaining only the presentative feature distributions and the LoRA blocks. Because the feature representation capability is similar for identical pre-trained LLMs. This allows for the expansion of any trained presentative feature distributions and LoRA blocks using the same model architecture on our proposed method simply by putting them together, regardless of who carried out the training. Notably, this expansion requires no additional training, meaning that the model gains associated knowledge simply by integrating the feature distributions and LoRA blocks. This expansion not only achieves plug-in continual learning but also promotes the development of the open-source community. We illustrate this flexible expansion in Figure 2, and we will further validate the effectiveness of the expansion in the subsequent experiments section.

## 4. Experiments

In this section, we provide extensive experiments to demonstrate the effectiveness of our proposed method.

Table 2: The test performance (%) on two continual learning benchmarks using the LLaMA-2-7B model. ✘ indicates methods with IL, while ✔ signifies methods without IL. All evaluation metrics are reported after the training on the last task of the benchmarks. In methods without IL, the best performances are highlighted in bold. The last column Avg. is the average of the performance on the two benchmarks.

| Methods | | SuperNI Benchmark | | | | Long Sequence Benchmark | | | | Avg.↑ |
|---|---|---|---|---|---|---|---|---|---|---|
| | | AP↑ | F.Ra↓ | FWT↑ | BWT↑ | AP↑ | F.Ra↓ | FWT↑ | BWT↑ | |
| Replay | ✘ | 39.48 | 14.86 | 0.19 | -26.47 | 71.43 | 13.64 | 0.97 | -12.73 | 55.46 |
| ProgPrompt | ✘ | 40.54 | 0.05 | -14.18 | 0.0 | 70.64 | 0.0 | -14.72 | 0.0 | 55.59 |
| SAPT-LoRA | ✘ | 56.23 | 1.07 | 0.81 | -0.65 | 81.75 | 2.81 | 1.09 | -2.53 | 68.99 |
| SeqLoRA | ✔ | 28.65 | 28.21 | -0.18 | -27.73 | 26.80 | 60.53 | -2.11 | -60.48 | 27.73 |
| ProgPrompt | ✔ | 14.34 | 28.12 | -14.18 | -28.07 | 10.67 | 64.25 | -14.72 | -64.25 | 12.51 |
| LFPT5 | ✔ | 38.71 | 16.81 | 0.32 | -15.42 | 70.31 | 5.63 | 0.51 | -4.32 | 54.51 |
| EPI | ✔ | - | - | - | - | 72.27 | 5.04 | -3.12 | -0.50 | - |
| O-LoRA | ✔ | 37.17 | 19.01 | 0.08 | -18.88 | 54.47 | 26.65 | -3.03 | -26.63 | 45.82 |
| SAPT-LoRA | ✔ | 30.28 | 26.97 | **0.74** | -26.97 | 7.08 | 82.15 | **0.05** | -82.12 | 18.68 |
| TASL-LoRA | ✔ | 42.00 | - | - | - | 75.00 | - | - | - | 58.50 |
| Ours-Dot | ✔ | 54.10 | 1.47 | 0.04 | -0.66 | 81.13 | 3.11 | -0.77 | -2.86 | 67.62 |
| Ours-L2 | ✔ | **55.60** | **0.23** | 0.08 | **-0.14** | **83.01** | **0.57** | -1.07 | **-0.41** | **69.31** |

## 4.1. Experimental Setup

**Datasets.** We conducted experiments on two CL benchmarks, including SuperNI Benchmark (Wang et al., 2022a) and Long Sequence Benchmark (Razdaibiedina et al., 2023).

The SuperNI Benchmark is a comprehensive benchmark of various NLP tasks, with instructions crafted by experts. It encompasses tasks like dialogue generation, information extraction, question answering, summarization, and sentiment analysis, and is used to evaluate the performance of the model in general scenarios. Following previous CL settings (Zhao et al., 2024), we selected three tasks from each category, creating a total of 15 task sequences for evaluation. For training, 1,000 instances are randomly sampled from the dataset for each task, with an additional 100 instances selected for validation and testing.

The Long Sequence Benchmark is a CL benchmark comprising 15 classification tasks. Following previous CL settings (Razdaibiedina et al., 2023; Wang et al., 2023a), we selected 1,000 random samples for training in each task and reserved 500 samples per category for validation and testing.

**Evaluation Metrics.** We denote the testing performance on the $j$-th task after training on the $i$-th task as $a_{ij}$, using Accuracy for classification tasks and Rouge-L (Lin, 2004) for other tasks. Following previous CL studies, we use widely adopted CL evaluation metrics: Average Performance (AP) (Chaudhry et al., 2018) evaluates the average performance of all tasks after training on the last task, i.e., $\frac{1}{K}\sum_{j=1}^{K} a_{Kj}$, which is the most important metric for evaluating CL methods; Forgetting Rate (F.Ra)

(Chaudhry et al., 2018) measures how much knowledge has been forgotten across the stream of incoming tasks, i.e., $\frac{1}{K-1}\sum_{j=1}^{K-1}(\max_{q=j}^{K-1} a_{qj} - a_{Kj})$; Forward Transfer (FWT) (Lopez-Paz & Ranzato, 2017) evaluates how much knowledge from previous tasks transfers to a new task, i.e., $\frac{1}{K}\sum_{j=1}^{K} a_{jj} - a_{0j}$, where $a_{0j}$ refers to the performance of training task $j$ individually; Backward Transfer (BWT) (Ke & Liu, 2022) measures how much learned tasks benefit from newly learned tasks, i.e., $\frac{1}{K-1}\sum_{j=1}^{K-1} a_{Kj} - a_{jj}$.

**Compared Methods.** We evaluate our method alongside the following PEFT-based CL methods: 1) Replay mitigates forgetting by periodically replaying samples from previously learned tasks, but it has the risk of IL; 2) SeqLoRA sequentially trains the LoRA on the task orders. 3) LFPT5 (Wang et al., 2022b) continuously trains a soft prompt for each task using generative replay and an auxiliary loss; 4) ProgPrompt (Razdaibiedina et al., 2023) trains a sequence of prompts in the order tasks arrive. Assumes the task-related ID is known during testing, allowing for direct selection of the corresponding prompt for evaluation. We evaluated the performance of ProgPrompt with and without IL. 5) EPI (Wang et al., 2023b) trains a prompt for each task and selects the prompt based on the distance between the input and the distributions of different classifier task labels; 6) O-LoRA (Wang et al., 2023a) ensures the newly learned LoRA components for each task are orthogonal to the previously learned ones to prevent interference with already acquired knowledge. 7) SAPT (Zhao et al., 2024) uses an attentive reflection module and a shared attention framework to adjust PEFT block learning and selection. However, the attentive

Table 3: The test performance (%) on two continual learning benchmarks using the LLaMA-2-13B model. ✘ indicates methods with IL, while ✔ signifies methods without IL. All evaluation metrics are reported after the training on the last task of the benchmarks. In methods without IL, the best performances are highlighted in bold. The last column Avg. is the average of the performance on the two benchmarks.

| Methods | | SuperNI Benchmark | | | | Long Sequence Benchmark | | | | Avg.↑ |
|---|---|---|---|---|---|---|---|---|---|---|
| | | AP↑ | F.Ra↓ | FWT↑ | BWT↑ | AP↑ | F.Ra↓ | FWT↑ | BWT↑ | |
| Replay | ✘ | 43.99 | 11.64 | 0.72 | -9.75 | 76.63 | 7.92 | 0.02 | -14.86 | 60.31 |
| ProgPrompt | ✘ | 38.93 | 0.35 | -17.35 | 0.0 | 72.43 | 0.0 | -14.39 | 0.0 | 55.68 |
| SAPT-LoRA | ✘ | 56.95 | 1.39 | 0.81 | -0.56 | 82.32 | 1.98 | 0.78 | -1.57 | 69.64 |
| SeqLoRA | ✔ | 30.07 | 28.50 | -0.03 | -28.06 | 50.76 | 35.96 | -2.65 | -35.80 | 40.42 |
| ProgPrompt | ✔ | 11.23 | 30.03 | -17.35 | -29.68 | 8.08 | 68.95 | -14.39 | -68.95 | 9.66 |
| LFPT5 | ✔ | 41.26 | 14.67 | -0.52 | -12.31 | 71.61 | 6.51 | -1.34 | -3.78 | 56.44 |
| EPI | ✔ | - | - | - | - | 76.66 | 4.91 | -0.09 | -1.03 | - |
| O-LoRA | ✔ | 44.49 | 12.28 | -0.65 | -11.94 | 57.34 | 28.23 | -3.13 | -28.22 | 50.92 |
| SAPT-LoRA | ✔ | 35.53 | 23.00 | 0.66 | -22.93 | 31.37 | 57.72 | **-0.05** | -57.72 | 33.45 |
| Ours-Dot | ✔ | 55.28 | 2.54 | **0.65** | -1.77 | 82.83 | 2.35 | -1.32 | -2.11 | 69.06 |
| Ours-L2 | ✔ | **56.47** | **0.27** | 0.05 | **0.16** | **84.75** | **0.39** | -0.98 | **-0.34** | **70.61** |

reflection module poses a risk of IL. 8) TaSL (Feng et al., 2024) identifies task-related areas through grouped skill localization and consolidates them to prevent forgetting. More complete experiments such as different models, different task orders and visualisations are provided in Appendix C.

**Implementation Details.** Following previous CL settings (Zhao et al., 2024), all methods are performed with instruction tuning (Wei et al., 2021; Ouyang et al., 2022) to leverage the task instruction provided in the two benchmarks. To validate the effectiveness of our proposed method, we considered pre-trained LLMs with two architectures: the encoder-decoder T5-large and the decoder-only LLaMA-2-7B and LLaMA-2-13B models, with model parameters ranging from 770M to 13B. To ensure a fair comparison, we use LoRA fine-tuning to train each task individually as $a_{0k}$ for all methods. For each transformer layer, we only count the feature distributions of Q and V components. For the temperature coefficient $T$ in the softmax function, we set all to 1.0 following previous studies (Zhao et al., 2024). For Top-$K$ activation in the dynamic selection module, we set $K$ to 1 In Table 2 and Table 3, while for the remainder of the experiments, $K$ was set to the total number of tasks. It is important to note that the values of $K$ during training and testing do not have to be identical. For simplicity, we chose to keep them the same. Detailed implementation details you can find in the Appendix B.

### 4.2. Experimental Results

**Performance on Continual Learning Benchmarks.** Table 2 and Tabel 3 show the comparison results of our proposed method and recent continual learning methods

for LLMs with PEFT (LoRA: SAPT-LoRA, SeqLoRA, O-LoRA, TASL; Prompt: ProgPrompt, LFPT5, EPI) on two popular CL benchmarks. We use the Top-1 strategy to select LoRA blocks. It can be observed that our method significantly outperforms other baselines without IL. For instance, in the SuperNI Benchmark, Ours-Dot and Ours-L2 achieve average performances of 54.10% and 55.60%, respectively, which are approximately 12.85% higher than the best method without IL, TaSL. Furthermore, compared to methods with IL, our proposed method achieves similar or even better CL performance. In the SuperNI Benchmark, our method is only 0.63% lower than the SOTA CL method with IL, SAPT-LoRA, while in the Long Sequence Benchmark, our method surpasses it by 1.26%. Moreover, in the comprehensive evaluation across the two benchmarks, our method establishes a new SOTA with 69.31% average performance. These results validate the effectiveness of our method for CL of LLMs. By comparing the experimental results in Table 2 and Table 3, it can be observed that the average performances of Ours-Dot and Ours-L2 using LLaMA-2-13B are 69.06% and 70.61%, respectively, both higher than the 67.62% and 69.31% achieved using LLaMA-2-7B. This indicates that more powerful models can achieve stronger CL performance.

**Performance of Zero-shot on Unseen Tasks.** Following previous studies (Zhao et al., 2024), we further select 3 tasks from five types of problems (Dialog, IE, QA, Sum, and SA) to evaluate the cross-task generalization ability of CL methods. Detailed information on task selection can be found in Appendix A.1. In Table 4, we can see the performance of our method and some CL methods on unseen

Table 4: The test performance (%) on the unseen task using the LLaMA-2-7B and LLaMA-2-13B model. ✘ indicates methods with IL, while ✔ signifies methods without IL. The best performances are highlighted in bold.

| Methods | | LLaMA-2-7B | | | | | | LLaMA-2-13B | | | | | |
| | | Dialog | IE | QA | Sum | SA | Avg. | Dialog | IE | QA | Sum | SA | Avg. |
| --- | --- | --- | --- | --- | --- | --- | --- | --- | --- | --- | --- | --- | --- |
| ProgPrompt | ✘/✔ | 7.16 | 6.65 | 17.69 | 9.86 | 4.34 | 9.14 | 7.92 | 6.36 | 16.91 | 14.64 | 9.0 | 10.97 |
| SAPT-LoRA | ✘ | 12.22 | 31.70 | 41.35 | 15.65 | 63.24 | 32.84 | 12.32 | 29.24 | 41.36 | 15.53 | 67.57 | 33.20 |
| SAPT-LoRA | ✔ | **12.42** | 34.31 | 49.11 | 18.24 | 58.33 | 34.48 | 11.84 | 35.89 | 47.52 | 19.80 | 72.16 | 37.45 |
| SeqLoRA | ✔ | 4.44 | 40.78 | 36.10 | 10.50 | 57.98 | 29.96 | 6.58 | 35.84 | 40.44 | 7.81 | 41.07 | 26.35 |
| O-LoRA | ✔ | 7.43 | 46.71 | 28.65 | 11.22 | 60.31 | 30.87 | 6.46 | 38.39 | 57.75 | 14.30 | 47.82 | 32.95 |
| Ours-Dot | ✔ | 9.52 | 40.55 | 49.94 | 21.32 | 59.57 | 36.18 | **12.37** | 32.91 | 55.27 | 20.32 | 65.31 | 37.24 |
| Ours-L2 | ✔ | 11.54 | **53.50** | **57.82** | **21.46** | **66.99** | **42.26** | 12.00 | **45.76** | **61.31** | **23.20** | 71.25 | **42.70** |

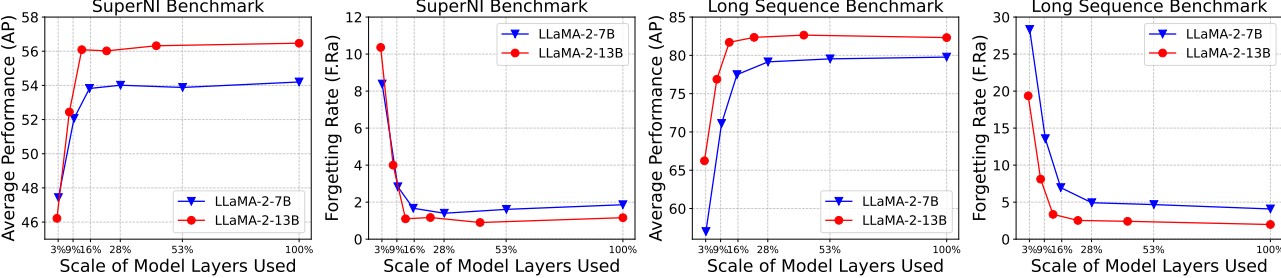

Figure 3: The performance (average performance ↑ and forgetting rate ↓) of our method for the LLaMA-2-7B and LLaMA-2-13B on SuperNI Benchmark and Long Sequence Benchmark as the number of model layers being used increases. Since the number of layers in different models varies, the scales are not strictly aligned.

tasks using LLaMA-2-7B and LLaMA-2-13B. According to these experimental results, we observe that our method achieves higher performance compared to other methods, whether using dot product similarity or L2 distance. This is attributed to the fact that our method uses a dynamic selection of modules to choose the appropriate knowledge (i.e., LoRA blocks) for each test instance. These results validate that our method possesses strong generalization capabilities to handle more realistic scenarios.

**Performance of Model Layer Usage.** For LoRA-based methods, we add LoRA blocks to the Query and Key components of the attention module at each layer. Consequently, our method requires analysis of the feature distribution in the Query and Key components at every layer. In Figure 3, we present the performance changes of our method, equipped with L2 distance, as the number of utilized layers increases. It can be observed that as more layers of the model are adopted, both the average performance and forgetting rate improve. Moreover, good results can be achieved with only a small number of layers being used. This indicates the potential of leveraging pre-trained models for feature extraction and validates the effectiveness of our method. In addition, our method only needs to increase the parameters for the LLaMA-2-7B model by 0.03% (including the LoRA

block) even with all layers, compared to 0.04% for SAPT.

**Comparison of Similarity Calculation.** In Figure 4, we present the results of two similarity calculation methods (L2 distance and dot product similarity) under different dynamic selection strategies (selecting the top-k most relevant). We observe that L2 distance consistently outperforms dot product similarity in terms of both performance and stability. This may be because the results of the dot product similarity are too close to each other. As shown in the visualization in Figure 1.(c) and 1.(d), while the dot product can identify the most relevant LoRA block, its values are relatively similar. As a result, the performance of the dot product similarity improves noticeably as top-k decreases. Moreover, our method surpasses the best baseline without DL in most cases, further validating the effectiveness of our method.

**Performance of Expansion.** In Table 5, we present the results of expansion feature distributions and LoRA blocks trained on two benchmarks. It is evident that models trained on a single benchmark can acquire knowledge when integrated with the presentative feature distributions and LoRA blocks trained on another benchmark, thanks to our dynamic selection module. Additionally, the combined performance of non-continual training is slightly worse than that of con-

Table 5: The test performance (%) on two continual learning benchmarks using the LLaMA-2-7B model and LLaMA-2-13B model with different LoRA blocks and feature distributions. "SuperNI" and "Long Sequence" indicate the average performance on the benchmarks using our method, where "+" denotes the expansion of LoRA blocks and presentative feature distributions as introduced in Section 3.5. "Continual" refers to each task within the benchmark being trained in a continual learning manner. "Non-Continual" refers to each task within the benchmark being trained independently rather than in a continual learning manner.

| LoRAs and Distributions | LLaMA-2-7B | | | LLaMA-2-13B | | |
| | SuperNI | Long Sequence | Avg. | SuperNI | Long Sequenc | Avg. |
| --- | --- | --- | --- | --- | --- | --- |
| Continual SuperNI
+ Non-Continual Long Sequence | 54.33
53.70 (-0.63) | 29.41
72.78 (+43.37) | 41.87
63.24 (+21.37) | 56.52
55.78 (-0.74) | 31.93
78.10 (+46.17) | 44.23
66.94 (+22.71) |
| Continual Long Sequence
+ Non-Continual SuperNI | 28.75
52.12 (+23.37) | 79.82
78.49 (-1.33) | 54.29
65.31 (+11.02) | 37.09
54.88 (+17.79) | 82.53
81.73 (-0.80) | 59.81
68.31 (+8.50) |
| Continual SuperNI + Long Sequence | 53.47 | 78.30 | 65.89 | 55.34 | 81.72 | 68.53 |

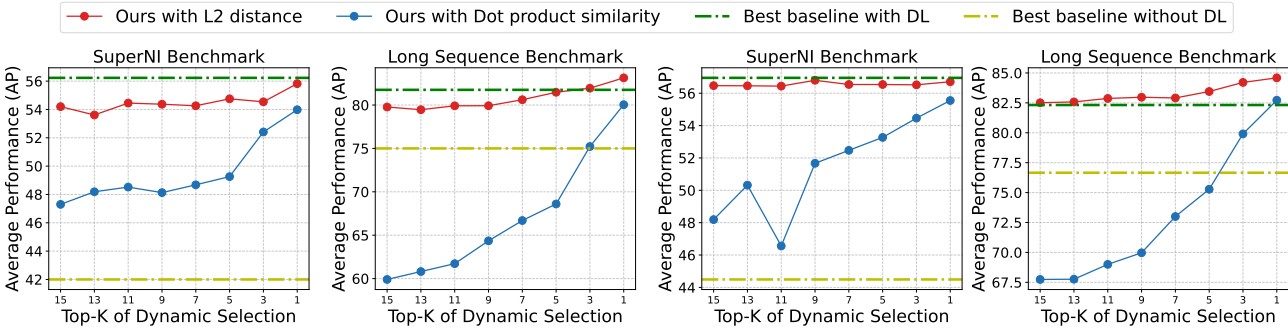

Figure 4: The performance (average performance ↑) of our method for the LLaMA-2-7B and LLaMA-2-13B on SuperNI Benchmark and Long Sequence Benchmark varies with the top-k selections of the dynamic selection module.

tinual training on two benchmarks separately and then combined. This is because the LoRA blocks trained independently do not have the capacity to utilize prior knowledge, which also validates the effectiveness of our method.

## 5. Conclusion

In this paper, we studied a critical problem in continual learning (CL) for large language models (LLMs) called Information Leakage (IL), which not only imposes potential risks on data privacy protection but also significantly hinders the deployment of LLMs in real-world online scenarios. In order to solve this problem, we first reconsidered the reasons for performance differences with and without IL in previous CL methods, mainly introducing new learnable parameters that brought about new forgetting. Second, we utilized well-developed pre-trained LLMs trained on large-scale data and proposed a novel online method exploiting presentative feature distributions for parameter-efficient continual learning of large language models. Finally, we provided an expansion to our method that allows LoRA blocks and presentative feature distributions using the same model architecture trained on different tasks to be integrated

directly without additional training, achieving plug-in CL for LLMs. Our method avoids IL while achieving outstanding CL performance. We hope that our in-depth study on IL in CL will inspire further research to consider this important problem, thereby facilitating the better deployment of LLMs with CL capabilities in real-world online scenarios.

## Acknowledgements

This research work is supported by the Big Data Computing Center of Southeast University. Xin Cheng was also supported by Alibaba Group through Research Intern Program. Feng Liu is supported by the Australian Research Council (ARC) with grant number DE240101089, LP240100101, DP230101540 and the NSF&CSIRO Responsible AI program with grant number 2303037.

## Impact Statement

This paper presents work whose goal is to advance the field of Machine Learning. There are many potential societal consequences of our work, none which we feel must be specifically highlighted here.

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

# A. Details of Benchmark

## A.1. SuperNI Benchmark

The Natural-Instructions project (Mishra et al., 2022; Wang et al., 2022a) is a high-quality NLP benchmark designed to assess a ability of model to generalize to unseen tasks. This generalization relies on understanding and reasoning based on natural language instructions, which describe a task clearly and comprehensively. A model capable of "understanding" language instructions can successfully tackle any unseen task once provided with the task instructions. Following previous setting (Zhao et al., 2024), we selected three tasks from each of five categories within the Natural-Instructions project: dialogue generation (Dialog) (Zhang, 2018; Zang et al., 2020; Peskov & Cheng, 2020), information extraction (IE) (Santus et al., 2015; Nye et al., 2018; Mostafazadeh et al., 2020), question answering (QA) (Dasigi et al., 2019; Talmor et al., 2018), summarization (Sum) (Narayan et al., 2018; Gliwa et al., 2019; Kim et al., 2019), and sentiment analysis (SA) (Socher et al., 2013; Saravia et al., 2018), creating a total of 15 tasks. For each task, we randomly selected 1,000 instances for training and 100 instances for evaluation and testing. In Table 6, we show the details of the 15 tasks that are selected. In addition, we show the details of several different streaming orders in table 7, and all our experiments in the main text use Order 1.

| Dataset name | Task | Metric |
|---|---|---|
| task639_multi_woz_user_utterance_generation | dialogue generation | Rouge-L |
| task1590_diplomacy_text_generation | dialogue generation | Rouge-L |
| task1729_personachat_generate_next | dialogue generation | Rouge-L |
| task181_outcome_extraction | information extraction | Rouge-L |
| task748_glucose_reverse_cause_event_detection | information extraction | Rouge-L |
| task1510_evalution_relation_extraction | information extraction | Rouge-L |
| task002_quoref_answer_generation | question answering | Rouge-L |
| task073_commonsenseqa_answer_generation | question answering | Rouge-L |
| task591_sciq_answer_generation | question answering | Rouge-L |
| task511_reddit_tifu_long_text_summarization | summarization | Rouge-L |
| task1290_xsum_summarization | summarization | Rouge-L |
| task1572_samsum_summary | summarization | Rouge-L |
| task363_sst2_polarity_classification | sentiment analysis | accuracy |
| task875_emotion_classification | sentiment analysis | accuracy |
| task1687_sentiment140_classification | sentiment analysis | accuracy |

Table 6: The details of selected 15 datasets in the SuperNI Benchmark

| Order | Task Sequence |
|---|---|
| 1 | task1572 → task363 → task1290 → task181 → task002 → task1510 → task639 → task1729 → task073 → task1590 → task748 → task511 → task591 → task1687 → task875 |
| 2 | task748 → task073 → task1590 → task639 → task1572 → task1687 → task591 → task363 → task1510 → task1729 → task181 → task511 → task002 → task1290 → task875 |

Table 7: The streaming orders of selected 15 datasets in the SuperNI Benchmark

In order to evaluate the performance of the learned model in terms of cross-task generalisation ability on the unseen task, we also selected three more tasks from the previous five types of problems; Dialog (Wei et al., 2018; Cho & May, 2020;

Aliannejadi et al., 2021), IE (Schmitz et al., 2012; Zlabinger et al., 2020; Nan et al., 2020), QA (Levy et al., 2017; Zhang et al., 2018; Min et al., 2020), Sum (Henderson et al., 2014; Syed et al., 2020; Hasan et al., 2021), and SA (Sheng & Uthus, 2020; Lowphansirikul et al., 2020) for evaluation. The details of the selected unseen tasks are provided in table 8.

| Dataset name | Task | Metric |
|---|---|---|
| task360_spolin_yesand_response_generation | Dialogue Generation | Rouge-L |
| task574_air_dialogue_sentence_generation | Dialogue Generation | Rouge-L |
| task1714_convai3_sentence_generation | Dialogue Generation | Rouge-L |
| task180_intervention_extraction | Information Extraction | Rouge-L |
| task678_ollie_actual_relationship_answer_generation | Information Extraction | Rouge-L |
| task1410_dart_relationship_extraction | Information Extraction | Rouge-L |
| task339_record_answer_generation | Question Answering | Rouge-L |
| task670_ambigqa_question_generation | Question Answering | Rouge-L |
| task1327_qa_zre_answer_generation_from_question | Question Answering | Rouge-L |
| task522_news_editorial_summary | Summarization | Rouge-L |
| task1356_xlsum_title_generation | Summarization | Rouge-L |
| task1499_dstc3_summarization | Summarization | Rouge-L |
| task421_persent_sentence_sentiment_classification | Sentiment Analysis | Accuracy |
| task833_poem_sentiment_classification | Sentiment Analysis | Accuracy |
| task929_products_reviews_classification | Sentiment Analysis | Accuracy |

Table 8: The details of 15 unseen datasets in the SuperNI Benchmark

## A.2. Long Sequence Benchmark

The Long Sequence benchmark is a benchmark constructed by selecting five tasks (AG News, Amazon reviews, Yelp reviews, DBpedia and Yahoo Answers) from the standard CL benchmark (Zhang et al., 2015), four tasks (MNLI, QQP, RTE, SST2) from the GLUE benchmark (Wang, 2018), five tasks (WiC, CB, COPA, MultiRC, BoolQ) from the SuperGLUE benchmark (Wang et al., 2019) and the IMDB movie reviews dataset (Maas et al., 2011).

| Dataset name | Category | Task | Domain | Metric |
|---|---|---|---|---|
| Yelp | CL Benchmark | Sentiment Analysis | Yelp reviews | Accuracy |
| Amazon | CL Benchmark | Sentiment Analysis | Amazon reviews | Accuracy |
| DBpedia | CL Benchmark | Topic Classification | Wikipedia | Accuracy |
| Yahoo | CL Benchmark | Topic Classification | Yahoo Q&A | Accuracy |
| AG News | CL Benchmark | Topic Classification | News | Accuracy |
| MNLI | GLUE | Natural Language Inference | Various | Accuracy |
| QQP | GLUE | Paragraph Detection | Quora | Accuracy |
| RTE | GLUE | Natural Language Inference | News, Wikipedia | Accuracy |
| SST-2 | GLUE | Sentiment Analysis | Movie Reviews | Accuracy |
| WiC | SuperGLUE | Word Sense Disambiguation | Lexical Databases | Accuracy |
| CB | SuperGLUE | Natural Language Inference | Various | Accuracy |
| COPA | SuperGLUE | Question and Answering | Blogs, Encyclopedia | Accuracy |
| BoolQA | SuperGLUE | Boolean Question and Answering | Wikipedia | Accuracy |
| MultiRC | SuperGLUE | Question and Answering | Various | Accuracy |
| IMDB | SuperGLUE | Sentiment Analysis | Movie Reviews | Accuracy |

Table 9: The details of selected 15 datasets in the Long Sequence Benchmark

To evaluate the stability of the method under different task orders, we provide two task orders in Table 10.

| Order | Task Sequence |
|---|---|
| 1 | mnli → cb → wic → copa → qqp → boolqa → rte → imdb → yelp → amazon → sst-2 → dbpedia → ag → multirc → yahoo |
| 2 | yelp → amazon → mnli → cb → copa → qqp → rte → imdb → sst-2 → dbpedia → ag → yahoo → multirc → boolqa → wic |

Table 10: The streaming orders of selected 15 datasets in the SuperNI Benchmark

| Top-K | | LLaMA-2-7B | | | | LLaMA-2-13B | | | |
|---|---|---|---|---|---|---|---|---|---|
| | | AP↑ | BWT↑ | F.Ra↓ | FWT↑ | AP↑ | F.Ra↓ | FWT↑ | BWT↑ |
| 1 | Dot | 80.04 | 4.43 | 0.14 | -4.16 | 82.72 | 2.47 | -0.56 | -2.14 |
| | L2 | 83.13 | 0.42 | -0.32 | -0.35 | 84.59 | 0.36 | -1.93 | -0.32 |
| 3 | Dot | 75.22 | 9.10 | -1.84 | -8.89 | 79.91 | 4.98 | -2.46 | -4.76 |
| | L2 | 81.94 | 1.87 | -0.40 | -1.54 | 84.21 | 0.65 | -2.05 | -5.88 |
| 5 | Dot | 68.60 | 16.22 | -1.82 | -16.01 | 75.28 | 9.73 | -2.56 | -9.62 |
| | L2 | 81.47 | 2.37 | -1.84 | -2.19 | 83.45 | 1.68 | -2.05 | -1.41 |
| 7 | Dot | 66.69 | 18.01 | -1.98 | -17.87 | 73.00 | 12.74 | -2.52 | -12.11 |
| | L2 | 80.61 | 3.32 | -1.82 | -3.13 | 82.91 | 2.24 | -2.15 | -1.87 |
| 9 | Dot | 64.35 | 20.55 | -1.95 | -20.42 | 69.97 | 15.52 | -2.48 | 15.38 |
| | L2 | 79.92 | 4.05 | -1.84 | -3.85 | 82.97 | 1.72 | -2.34 | -1.62 |
| 11 | Dot | 61.72 | 23.46 | -1.90 | -23.28 | 69.00 | 16.74 | -2.38 | -16.54 |
| | L2 | 79.91 | 4.13 | -1.80 | -3.91 | 82.87 | 2.29 | -1.94 | -2.16 |
| 13 | Dot | 60.81 | 24.40 | -1.92 | 24.24 | 67.76 | 18.09 | -2.36 | -17.89 |
| | L2 | 79.46 | 4.63 | -1.79 | -4.40 | 82.58 | 2.33 | -2.28 | -2.10 |
| 15 | Dot | 59.90 | 25.33 | -0.42 | -25.13 | 67.74 | 17.85 | -1.01 | -17.71 |
| | L2 | 79.77 | 4.08 | -0.42 | -3.84 | 82.51 | 2.52 | -2.17 | 2.29 |

Table 11: The test performance (%) on the Long Sequence Benchmark using the LLaMA-2-7B and LLaMA-2-13B model with different top-k selections of the dynamic selection module. All evaluation metrics are reported after the training on the last task of the benchmarks.

## B. Implementation Details

For our proposed method, we utilize the AdamW optimizer to train the models. Specifically, we employ a learning rate of 3e-4 for the T5-Large model, and 5e-5 for both the LLaMA-2-7B and LLaMA-2-13B models. For each transformer layer, we only count the feature distributions of $Q$ and $V$ components. For the T5-Large model, we only consider the statistical encoder.

The T5-Large model is trained on a single NVIDIA Tesla A100 GPU, with a batch size of 32 and a low-rank parameter $r$ set to 8. We train the model 100 epochs on the SuperNI benchmark and 10 epochs on the Long Sequence benchmarks.

For the LLaMA-2-7B and LLaMA-2-13B models, training is conducted on 8 NVIDIA Tesla A100 GPUs utilizing the DeepSpeed repository. The total batch size is 32, and both benchmarks use a low-rank parameter $r$ of 4. We train the model 50 epochs on the SuperNI benchmark and 20 epochs on the Long Sequence benchmarks.

The primary experiments reported in the paper are the averages obtained from two different orders in Table 10, while other experiments showcase results based on order 1.

For the temperature coefficient $T$ in the dynamic selection module of our method, we set all to 1.0. For the methods we compared, we reimplemented their method based on the open source code of SAPT (Zhao et al., 2024), which makes their results potentially better than what they reported in their original paper. For TASL (Feng et al., 2024), we directly extracted

| Methods | Order | SuperNI Benchmark | | | | Long Sequence Benchmark | | | |
|---------|-------|------|------|------|------|------|------|------|------|
| | | AP↑ | BWT↑ | F.Ra↓ | FWT↑ | AP↑ | F.Ra↓ | FWT↑ | BWT↑ |
| Ours-Dot | 1 | 53.98 | 1.52 | -0.06 | -0.72 | 80.04 | 4.43 | 0.14 | -4.16 |
| | 2 | 54.30 | 1.42 | 0.14 | -0.60 | 82.22 | 1.78 | -1.68 | -1.56 |
| Ours-L2 | 1 | 55.82 | 0.22 | 0.27 | -0.17 | 83.13 | 0.42 | -0.32 | -0.35 |
| | 2 | 55.38 | 0.24 | -0.12 | -0.22 | 82.89 | 0.72 | -1.83 | -0.47 |

Table 12: The test performance (%) on the Long Sequence Benchmark using the LLaMA-2-7B model with different task order. All evaluation metrics are reported after the training on the last task of the benchmarks.

| Methods | Order | SuperNI Benchmark | | | | Long Sequence Benchmark | | | |
|---------|-------|------|------|------|------|------|------|------|------|
| | | AP↑ | BWT↑ | F.Ra↓ | FWT↑ | AP↑ | F.Ra↓ | FWT↑ | BWT↑ |
| Ours-Dot | 1 | 55.55 | 2.56 | 0.73 | -1.56 | 82.72 | 2.47 | -0.56 | -2.14 |
| | 2 | 55.01 | 2.51 | 0.56 | -1.97 | 82.93 | 2.22 | -2.07 | -1.93 |
| Ours-L2 | 1 | 56.71 | 0.05 | 0.13 | 0.33 | 84.59 | 0.36 | -0.39 | -0.32 |
| | 2 | 56.22 | 0.49 | -0.03 | -0.02 | 84.91 | 0.42 | -1.57 | -0.36 |

Table 13: The test performance (%) on the Long Sequence Benchmark using the LLaMA-2-13B model with different task order. All evaluation metrics are reported after the training on the last task of the benchmarks.

the results they reported in their paper.

## C. Additional Results

### C.1. Complete Results on LLaMA-2-7B

For LLaMA-2-7B model, the main experiments are provided in Table 2. In order to verify the effectiveness of our method under different task orders, we provide the results of our method under two orders in Table 12. It can be seen that there is no significant difference in our method under different orders, which indicates the stability of our method.

### C.2. Complete Results on LLaMA-2-13B

In Table 3, we show the comparison results of our method and recent continual learning methods for LLMs with PEFT (LoRA: SAPT-LoRA, SeqLoRA, O-LoRA; Prompt: Prog-Prompt, LFPT5, EPI) on two popular CL benchmarks, using the LLaMA-2-13B model. Our method uses a Top-1 strategy to select LoRA blocks. As illustrated in the table, our method consistently outperforms other baselines without IL, regardless of the similarity method used. Notably, when using L2 distance, our method achieves the best average performance across these two benchmarks, which is the same as what we found in the main text. We also show in Table 13 how our method performs under different task orders, and these results again demonstrate the effectiveness of our method.

### C.3. Complete Results on T5-Large

In Table 14, we show the comparison results of our method and recent continual learning methods for LLMs with PEFT (LoRA: SAPT-LoRA, SeqLoRA, O-LoRA, TASL-LoRA; Prompt: Prog-Prompt, LFPT5, EPI, L2P) on two popular CL benchmarks, using the T5-Large model. Our method uses a Top-1 strategy to select LoRA blocks. As illustrated in the table, our method achieves good performance on the SuperNI Benchmark over other baselines without IL and close to the performance of the method with IL. On the Long Sequence Benchmark, although there is not much difference with other methods without IL, the overall average performance is still the best without IL, which also validates the effectiveness of our method in different frameworks.

| Methods | | SuperNI Benchmark | | | | Long Sequence Benchmark | | | | Avg.↑ |
|---|---|---|---|---|---|---|---|---|---|---|
| | | AP↑ | F.Ra↓ | FWT↑ | BWT↑ | AP↑ | F.Ra↓ | FWT↑ | BWT↑ | |
| Replay | ✘ | 35.37 | 16.92 | -1.35 | -15.79 | 71.28 | 13.05 | 1.28 | -12.18 | 53.33 |
| SAPT-LoRA | ✘ | 51.50 | 0.91 | 1.88 | -0.57 | 82.02 | 1.50 | 1.86 | 1.25 | 66.76 |
| TASL-LoRA | ✘ | 53.01 | - | - | -0.81 | 84.33 | - | - | -0.98 | 68.67 |
| SeqLoRA | ✔ | 6.43 | 33.39 | -13.58 | -30.94 | 9.72 | 78.61 | 0.81 | -73.37 | 8.08 |
| L2P | ✔ | 12.73 | 11.87 | -19.14 | -7.95 | 57.98 | 22.49 | 1.36 | -16.63 | 35.36 |
| ProgPrompt | ✔ | 3.34 | 35.57 | -3.29 | -33.18 | 7.98 | 71.55 | -2.63 | -66.71 | 5.66 |
| LFPT5 | ✔ | 34.37 | 15.80 | -0.46 | -14.47 | 67.01 | 13.89 | **2.48** | -12.80 | 50.69 |
| EPI | ✔ | - | - | - | - | **75.15** | **1.61** | -0.77 | **-1.42** | - |
| O-LoRA | ✔ | 25.89 | 26.37 | -0.14 | -24.59 | 69.24 | 7.00 | -8.15 | -4.05 | 47.57 |
| SAPT-LoRA | ✔ | 11.12 | 42.83 | **0.70** | -40.44 | 10.18 | 78.45 | 1.93 | -73.22 | 10.65 |
| TASL-LoRA | ✔ | 27.43 | - | - | -16.91 | 72.29 | - | - | -2.04 | 49.86 |
| Ours-Dot | ✔ | 49.25 | 2.89 | -0.43 | -2.56 | 70.50 | 9.61 | 1.86 | -9.53 | 59.88 |
| Ours-L2 | ✔ | **51.88** | **0.48** | 0.05 | **-0.26** | 70.05 | 9.61 | 1.87 | -9.53 | **60.97** |

Table 14: The test performance (%) on two continual learning benchmarks using the T5-Large model. ✘ indicates methods with IL, while ✔ signifies methods without IL. All evaluation metrics are reported after the training on the last task of the benchmarks. In methods without IL, the best performances are highlighted in bold. The last column Avg. is the average of the performance on the two benchmarks.

| $T$ | SuperNI Benchmark | | | | Long Sequence Benchmark | | | | Avg.↑ |
|---|---|---|---|---|---|---|---|---|---|
| | AP↑ | F.Ra↓ | FWT↑ | BWT↑ | AP↑ | F.Ra↓ | FWT↑ | BWT↑ | |
| 1.0 | 54.20 | 1.86 | **0.66** | -1.26 | 79.77 | 4.08 | **-0.42** | -3.84 | 66.99 |
| 0.6 | 53.99 | 1.30 | -0.01 | -0.76 | 81.67 | 2.36 | -1.62 | -2.21 | 67.83 |
| 0.2 | **54.75** | **0.58** | 0.43 | **-0.42** | **83.06** | **0.95** | -1.74 | **-0.59** | **68.91** |

Table 15: The test performance (%) on two continual learning benchmarks with temperature $T$ changes using the LLaMA-2-7B model equipped L2 distance. All evaluation metrics are reported after the training on the last task of the benchmarks with Order 1. The last column Avg. is the average of the performance on the two benchmarks.

| $T$ | SuperNI Benchmark | | | | Long Sequence Benchmark | | | | Avg.↑ |
|---|---|---|---|---|---|---|---|---|---|
| | AP↑ | F.Ra↓ | FWT↑ | BWT↑ | AP↑ | F.Ra↓ | FWT↑ | BWT↑ | |
| 1.0 | 56.47 | 1.16 | **0.52** | -0.36 | 82.51 | 2.52 | -2.17 | **2.29** | 69.49 |
| 0.6 | 56.67 | 0.92 | 0.36 | 0.03 | 83.81 | 1.10 | -2.08 | -0.98 | 70.24 |
| 0.2 | **56.96** | **0.29** | 0.46 | **0.24** | **84.67** | **0.64** | -1.74 | -0.43 | **70.82** |

Table 16: The test performance (%) on two continual learning benchmarks with temperature $T$ changes using the LLaMA-2-13B model equipped L2 distance. All evaluation metrics are reported after the training on the last task of the benchmarks with Order 1. The last column Avg. is the average of the performance on the two benchmarks.

## C.4. Hyperparameter

In the dynamic selection module, we use the softmax function that includes a inherent temperature hyperparameter $T$. Following previous studies (Zhao et al., 2024), we fix this coefficient to 1.0. We add additional experiments concerning the impact of the temperature coefficient in Table 15 and Table 16. In fact, the temperature coefficient and the $K$ have similar usage, as they both adjust the scaling of weights. When the $T$ methodes 0, it effectively corresponds to setting $K$ to 1. Therefore, it is not necessary to simultaneously adjust multiple parameters externally. We can only adjust one of them. In addition, we provide experimental results for different LoRA ranks in Table 17.

| $d$ | SuperNI Benchmark | | | | Long Sequence Benchmark | | | | Avg.↑ |
|---|---|---|---|---|---|---|---|---|---|
| | AP↑ | F.Ra↓ | FWT↑ | BWT↑ | AP↑ | F.Ra↓ | FWT↑ | BWT↑ | |
| 2 | 54.03 | 1.23 | -0.63 | **-0.85** | **79.84** | 4.46 | -1.77 | -4.01 | 66.94 |
| 4 | 54.20 | 1.86 | 0.66 | -1.26 | 79.77 | **4.08** | **-0.42** | **-3.84** | 66.99 |
| 6 | **54.49** | **1.40** | **0.77** | -1.06 | 79.75 | 4.36 | -1.87 | -4.00 | **67.12** |

Table 17: The test performance (%) on two continual learning benchmarks with LoRA rank $d$ changes using the LLaMA-2-7B model equipped L2 distance. All evaluation metrics are reported after the training on the last task of the benchmarks with Order 1. The last column Avg. is the average of the performance on the two benchmarks.

| Methods | Memory↓ | Training Time↓ |
|---|---|---|
| O-LoRA | **2.097M (0.03%)** | 37:04 |
| Ours | 2.359M (0.03%) | **10:47** |
| SAPT | 2.928M (0.04%) | 13:52 |

Table 18: The report on memory usage (for LLaMA-7B, rank=4) and training time (on the last task in Long Sequence benchmark) for the three methods.

## C.5. Similarity Measurement

In the main text, we discussed two similarity measurement methods: Negative $L2$ euclidean distance and dot product similarity. We also considered the results of using cosine similarity for similarity measurement in Table 19 and Table 20. Actually, this also means that our proposed method does not rely on a specific similarity measure, which makes our method more flexible with plug-in similarity measures.

## C.6. Memory and time

In Table 18, we provide a comparison of memory and training time. While our method stores additional feature distributions compared with O-LoRA, these distributions are represented as lightweight vectors. Even when stored across all layers, they introduce only 0.262M additional parameters. As shown in Figure 3, retaining feature distributions for only a subset of layers achieves strong CL performance while further reducing memory overhead.

SAPT and our method incur only a small increase in lightweight computations during training. However, the parameters added by SAPT require gradient updates and replay data, resulting in slower training speed for SAPT. In contrast, O-LoRA necessitates calculating the square difference between LoRA parameters during loss computation, which introduces substantial computational overhead and leads to the slowest training speed.

## C.7. Additional dynamic selection experiments

We provide the average performance of our method under different Top-k strategies for dynamic selection in Figure 4. In order to provide details on other metrics, we provide detailed experimental results in Table 11. As the Top-K value decreases, we observe noticeable improvements across various metrics, indicating that our method effectively identifies the most relevant LoRA blocks. It is important to note that selecting the Top-1 does not imply that our method relies on only a single LoRA block for prediction. This is because our method is implemented at each transformer layer, allowing for different selections across layers. Therefore, even when using the Top-1 selection, our method fully leverages the knowledge from different LoRA blocks.

## C.8. Visualisation experiments

In Figure 1, we present visualizations of the LoRA block selection results, which are averaged across all layers. To offer a more detailed view, we further provide the results for each individual layer in Figure 5, Figure 6, Figure 7 and Figure 8. We can observe that in the shallow layers, there are no particularly prominent weights, indicating that the presentative feature distributions in the earlier layers are relatively similar, and the model is likely capturing more superficial features. As the

| Methods | SuperNI Benchmark | | | | Long Sequence Benchmark | | | | Avg.↑ |
|---|---|---|---|---|---|---|---|---|---|
| | AP↑ | F.Ra↓ | FWT↑ | BWT↑ | AP↑ | F.Ra↓ | FWT↑ | BWT↑ | |
| Ours-Cosine | 54.29 | 0.98 | **0.34** | -0.82 | 83.02 | 0.83 | -1.78 | -0.59 | 68.66 |
| Ours-Dot | 53.98 | 1.52 | -0.06 | -0.72 | 80.04 | 4.43 | **0.14** | -4.16 | 67.01 |
| Ours-L2 | **55.82** | **0.22** | 0.27 | **-0.17** | **83.13** | **0.42** | -0.32 | **-0.35** | **69.48** |

Table 19: The test performance (%) on two continual learning benchmarks (Top-$k$=1) using the LLaMA-2-7B model. All evaluation metrics are reported after the training on the last task of the benchmarks with **Order 1**. The last column Avg. is the average of the performance on the two benchmarks.

| Methods | SuperNI Benchmark | | | | Long Sequence Benchmark | | | | Avg.↑ |
|---|---|---|---|---|---|---|---|---|---|
| | AP↑ | F.Ra↓ | FWT↑ | BWT↑ | AP↑ | F.Ra↓ | FWT↑ | BWT↑ | |
| Ours-Cosine | 56.65 | 0.94 | 0.44 | -0.7 | 84.50 | 0.43 | -1.71 | **-0.32** | 70.58 |
| Ours-Dot | 55.55 | 2.56 | **0.73** | -1.56 | 82.72 | 2.47 | -0.56 | -2.14 | 69.14 |
| Ours-L2 | **56.71** | **0.05** | 0.13 | **0.33** | **84.59** | **0.36** | **-0.39** | **-0.32** | **70.65** |

Table 20: The test performance (%) on two continual learning benchmarks (Top-$k$=1) using the LLaMA-2-13B model. All evaluation metrics are reported after the training on the last task of the benchmarks with **Order 1**. The last column Avg. is the average of the performance on the two benchmarks.

layers deepen, there is a clear tendency in the weights, with the model capturing distinctive features based on characteristics, whether they be in $Q$ or $V$. Although the trends of $Q$ and $V$ are the same, there are still differences between them. For example, Q seems to have larger weights compared to V, but both are able to identify the most relevant LoRA blocks. These visualization results further validate the effectiveness of our method.

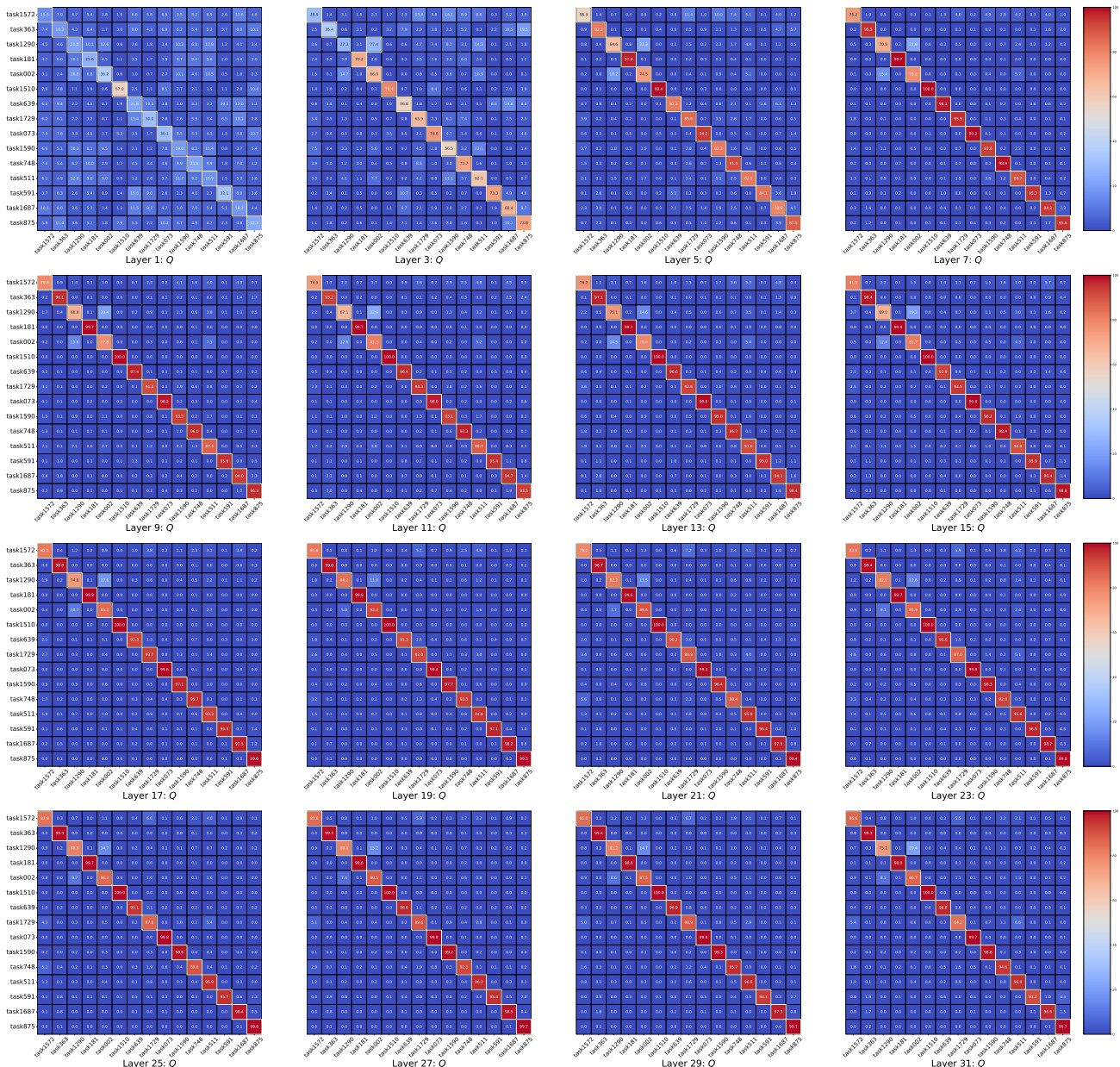

Figure 5: The heatmap shows the selection of LoRAs for each layer $Q$ of our method equipped with the L2 distance in the 15 tasks of the SuperNI benchmark. For all results, we scaled the values by a factor of 100 and highlighted the highest value in each row with a yellow box.

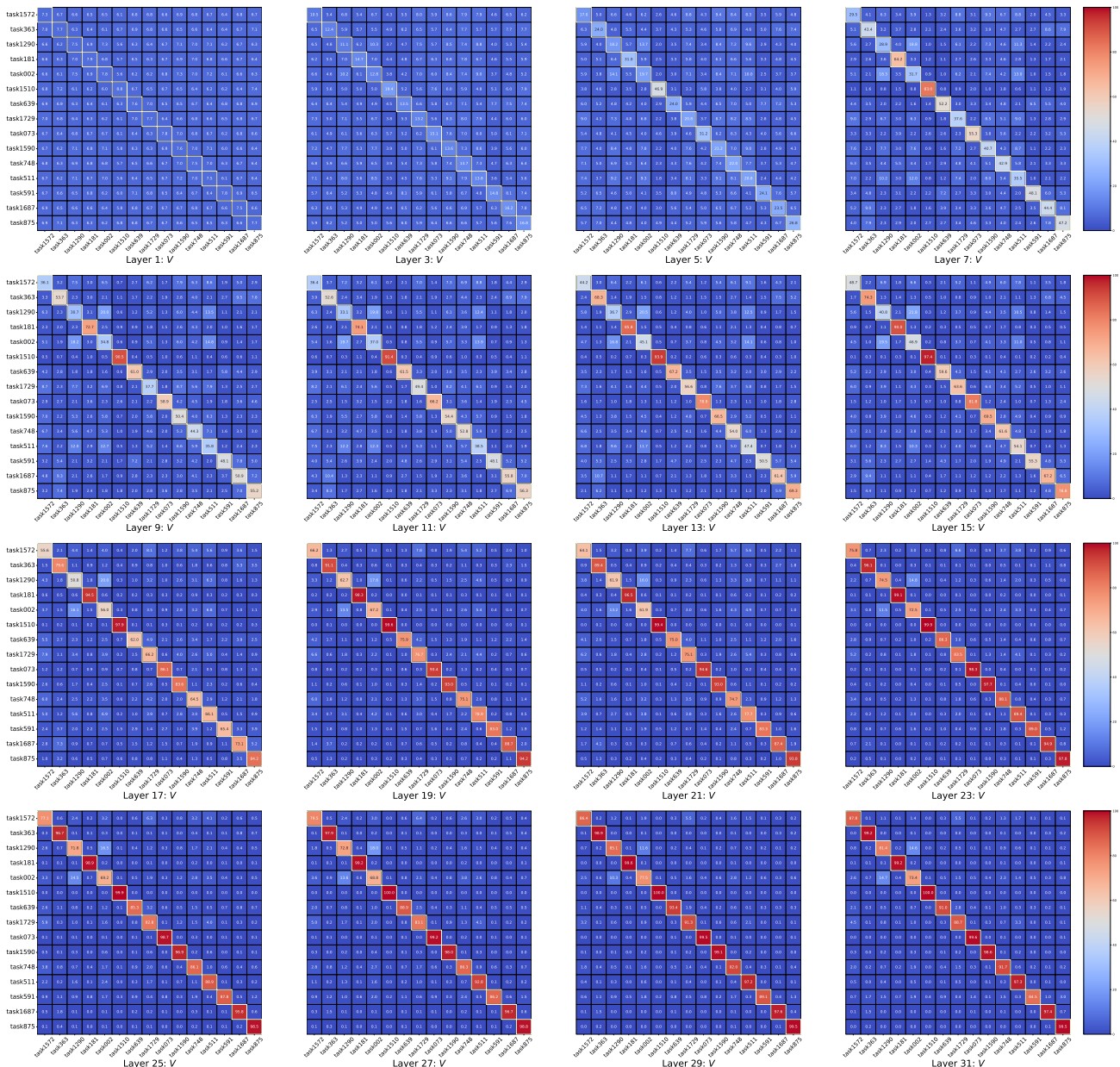

Figure 6: The heatmap shows the selection of LoRAs for each layer $V$ of our method equipped with the L2 distance in the 15 tasks of the SuperNI benchmark. For all results, we scaled the values by a factor of 100 and highlighted the highest value in each row with a yellow box.

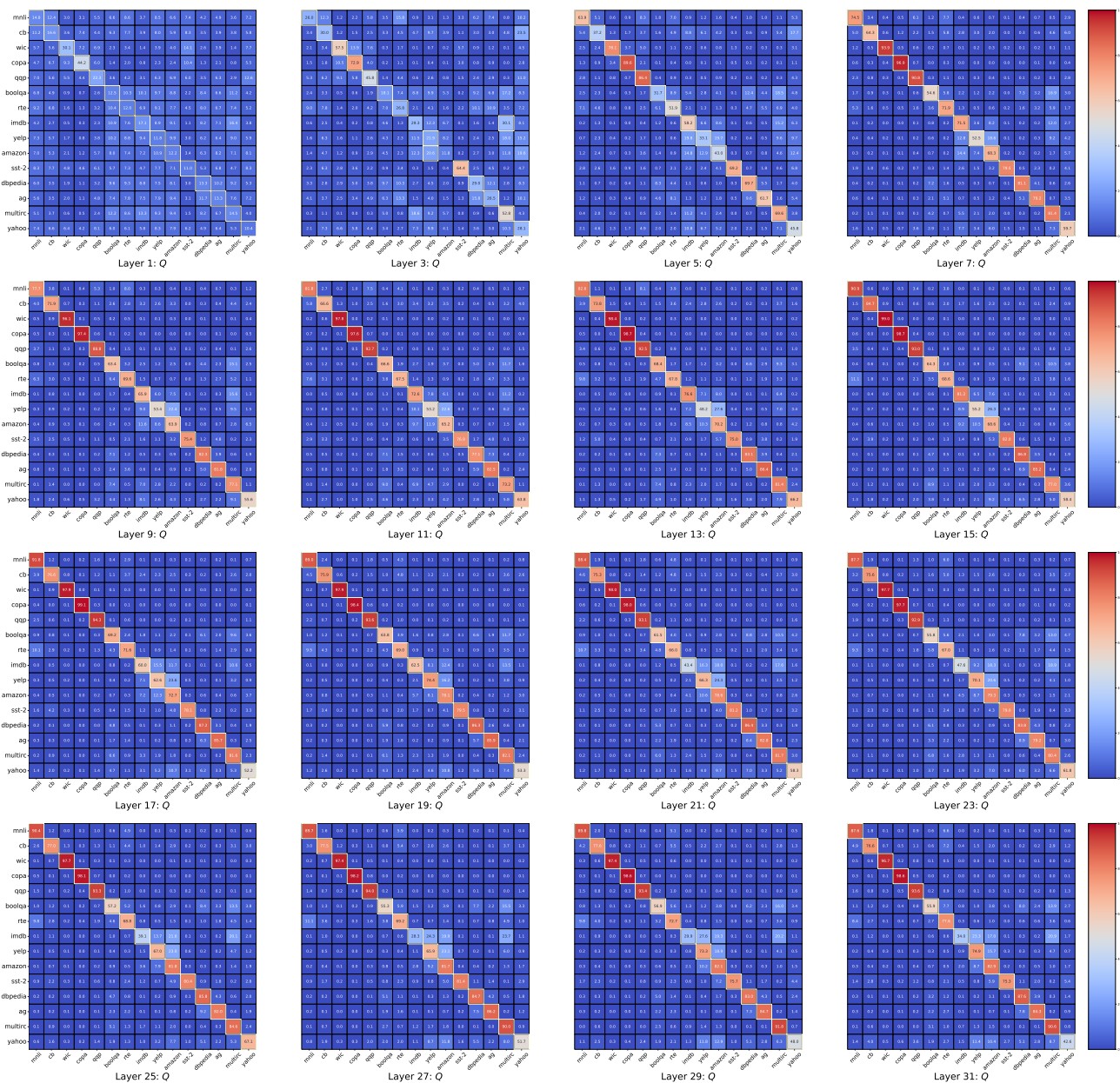

Figure 7: The heatmap shows the selection of LoRAs for each layer $Q$ of our method equipped with the L2 distance in the 15 tasks of the Long Sequence Benchmark. For all results, we scaled the values by a factor of 100 and highlighted the highest value in each row with a yellow box.

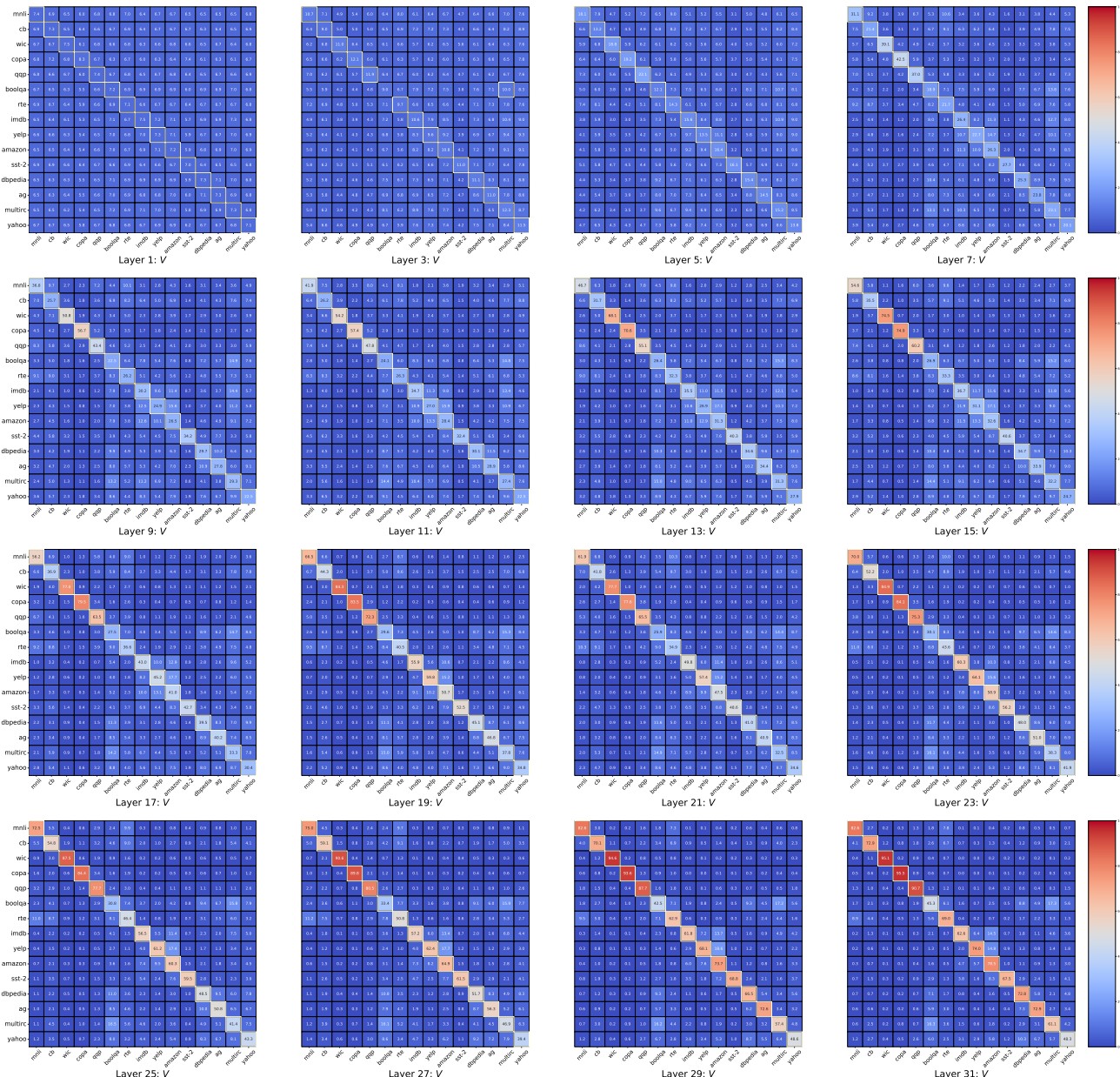

Figure 8: The heatmap shows the selection of LoRAs for each layer $V$ of our method equipped with the L2 distance in the 15 tasks of the Long Sequence Benchmark. For all results, we scaled the values by a factor of 100 and highlighted the highest value in each row with a yellow box.

