# OpenReview forum: "Exploiting Presentative Feature Distributions for Parameter-Efficient Continual Learning of Large Language Models"
_ICML.cc/2025/Conference — ICML 2025 poster_

### Official Review · Reviewer_54sH · 2025-03-12

**Overall Recommendation:** 4

**Summary:**

This paper proposes a novel parameter-efficient continual learning (CL) framework for large language models (LLMs) that leverages pre-trained model representations to dynamically select task-specific LoRA blocks via presentative feature distributions. The method addresses the critical challenge of information leakage (IL) inherent in prior CL approaches (e.g., data replay or parameter isolation with shared modules) by eliminating dependency on historical task data or identifiers during inference. Experimental validation on the SuperNI and Long Sequence benchmarks demonstrates state-of-the-art performance among IL-avoidant methods while maintaining competitive results compared to methods with IL risks.

**Claims And Evidence:**

Yes, the claims are well-supported by empirical evidence.

**Essential References Not Discussed:**

I am not aware of a crucial missing reference that is essential for contextualizing the approach.

**Experimental Designs Or Analyses:**

The experiments are well-structured: they compare multiple baselines under the same conditions (model, tasks, orders), and they measure standard metrics for CL.

**Methods And Evaluation Criteria:**

Yes.

**Other Comments Or Suggestions:**

A more detailed analysis of inference speed and memory usage would help clarify real deployment feasibility.

**Other Strengths And Weaknesses:**

Strengths:
1. The proposed framework contains presentative feature distribution and dynamic similarity-based selection which avoids common issues such as additional forgetting from new parameters or information leakage from data replay.
2. The method addresses critical real-world needs—such as high training costs for LLMs, model scalability, and privacy constraints that prevent historical data reuse—by enabling data-free (or replay-free) continual learning with robust performance.
3. Comprehensive experimental results on two representative benchmarks (SuperNI and Long Sequence) demonstrate the effectiveness of the proposed method.
Weakness:
1.While similarity metrics (e.g., L2 distance, dot product) are explained, the theoretical rationale for using high-dimensional feature distributions to stably represent task-specific domains remains underdeveloped. Further analysis of theoretical robustness and consistency in complex scenarios can enhance the quality of the paper.
2.The fixed or limited tuning of hyperparameters (e.g., temperature coefficient , LoRA rank) raises questions about their adaptability to varying data scales or task distributions. Systematic hyperparameter search or adaptive strategies need further exploration.

**Questions For Authors:**

1.How does the method disambiguate tasks with overlapping feature distributions (e.g., sentiment analysis across domains)? Does layer-wise selection (Figures 5–8) inherently mitigate this?
2.What is the maximum K tested? How does the quadratic similarity computation (for K tasks) impact real-time inference?
3.While similarity metrics (e.g., L2 distance, dot product) are explained, the theoretical rationale for using high-dimensional feature distributions to stably represent task-specific domains remains underdeveloped. Further analysis of theoretical robustness and consistency in complex scenarios can enhance the quality of the paper.
4.The fixed or limited tuning of hyperparameters (e.g., temperature coefficient , LoRA rank) raises questions about their adaptability to varying data scales or task distributions. Systematic hyperparameter search or adaptive strategies need further exploration.

**Relation To Broader Scientific Literature:**

This paper builds on parameter-efficient fine-tuning (PEFT) techniques like LoRA, and addresses the continual learning problem which is well-studied in classical settings. They also link to recent LLM-based CL methods such as SAPT, O-LoRA, and prompt-based methods.

**Theoretical Claims:**

I did not find significant formal proofs to check for correctness; it is primarily an algorithmic and empirical contribution.

---

> ### Author Rebuttal · Authors · 2025-03-31
>
> Thank you for recognizing our work and taking the time to review our manuscript. Below is our response to address your concerns.
>
> **Q1: (Questions) How does the method disambiguate ...... layer-wise selection inherently mitigate this?**
>
> A1: Our method can effectively identify tasks with overlapping feature distributions. In cases where we have two identical tasks, the dynamic selection process will allocate selections evenly between them, similar to how each task would be trained independently. Furthermore, the layer-wise selection process offers additional mitigation against overlapping feature distributions. As evidenced by our visualizations in Figures 5–8, the model learns progressively from shallow to deep layers, and our method can capture and distinguish the nuances of feature overlaps.
>
> **Q2: (Questions) What is the maximum K tested? How ...... computation impact real-time inference?**
>
> A2: The parameter $K$ used for testing was consistent with that used during training. Specifically, in Tables 2, 10, and 14, we set $K$ to 1, while for the remainder of the experiments, $K$ was set to the total number of tasks. It is important to note that the values of $K$ during training and testing do not have to be identical. For simplicity, we chose to keep them the same.
>
> During real-time inference, our method calculates the similarity between instances and the stored feature distributions and uses this similarity to select PEFT blocks. This process only adds a step for similarity computation compared to the original inference. Since quadratic similarity, dot-product similarity, and cosine similarity are easily implemented with matrix parallelization, the real-time inference time does not significantly change with different $K$.
>
> As reflected in Figure 4, we present the results of quadratic similarity under different $K$. It can be observed that quadratic similarity maintains stable performance across various $K$ values. This stability arises because quadratic similarity is unbounded before normalization, leading to more extreme weights after normalization. Different values of $K$ prioritize the exclusion of irrelevant LoRA blocks, and these blocks inherently have low weights when using quadratic similarity.
>
> **Q3: (Questions) While similarity metrics are explained ...... enhance the quality of the paper.**
>
> A3: We agree that a stronger theoretical basis would enhance the quality of this paper. However, theoretical frameworks related to large models, particularly in high-dimensional spaces such as hidden layers and feature spaces, are still under development. **All existing research lacks theoretical assurance.** Fortunately, we can provide some theoretical support for our method.
>
> Inspired by the studies on model merging[1], changes in model parameters are considered as directions that contain knowledge of specific tasks, referred to as task vectors. Since LoRA blocks are ultimately added to the original model parameters in our method, each  LoRA blocks $A_k$ and $B_k$ can also be seen as task vectors, offering rationale for our dynamic selection.
>
> For any given instance feature $\mathbf{W}^{l}h^{l}(x)$, it has undergone the same pre-trained parameters $\mathbf{W}^{l}$, thereby being a linear transformation applied uniformly across all instances. Given that related tasks require related knowledge, the final output space should be similar (e.g., vocabulary or patterns). Therefore, we establish a feature distribution $ D^{l} _{k} = \mathbb{E} _{p(x^{k}, y^{k})} [\mathbf{W}^{l}h^{l}(x^{k})] $ for the transformed features, and select blocks based on the distance to these distributions.
>
> [1] Gabriel Ilharco et al. Editing Models with Task Arithmetic. In NeurIPS, 2023.
>
> **Q4: (Questions) The fixed or limited tuning of hyperparameters ...... need further exploration.**
>
> A4: The temperature coefficient is an inherent hyperparameter of the softmax function, and the LoRA rank is inherent to LoRA fine-tuning. These two hyperparameters are not unique to our method. In Figure 4, we have discussed the specific hyperparameter $K$ associated with our method. To further validate the adaptability of our method across different hyperparameter settings, we have conducted additional experiments in the anonymous link https://anonymous.4open.science/r/ICML-2025-Rebuttal-10369/ICML_2025_Rebuttal.pdf (Table 20, 21 and 22).
>
> In fact, the temperature coefficient and the parameter $K$ serve similar purposes, as they both adjust the scaling of weights. When the temperature $T$ approaches 0, it effectively corresponds to setting K to 1. Therefore, it is not necessary to simultaneously adjust multiple parameters externally, just choose one to adjust.
>
> We agree that exploring adaptive hyperparameter tuning and systematic search is a promising area of research. This issue remains challenging, especially in large models, which have numerous hyperparameters that can be adjusted. We will make this challenging problem our future work.

---

> > ### Comment · Reviewer_54sH · 2025-04-07
> >
> > Thanks for the reply. I have no further questions and decide to keep my rating.

---

> > > ### Author Response · Authors · 2025-04-07
> > >
> > > Thank you so much for checking our rebuttal and acknowledging that you do not have any other concerns. We sincerely appreciate your positive feedback on our work!

---

### Official Review · Reviewer_MixK · 2025-03-13

**Overall Recommendation:** 2

**Summary:**

This paper presents a novel continual learning method for LLMs that avoids information leakage by employing presentative feature distributions. The proposed method characterizes parameter-efficient fine-tuning blocks using feature distributions and dynamically selects suitable blocks based on similarity metrics. The proposed methods perform well in continual learning benchmarks without accessing previous data. Extensive experiments on SuperNI and Long Sequence benchmarks validate its effectiveness.

**Claims And Evidence:**

The claims made in the submission are supported by empirical evidence and supportive experiments.

**Essential References Not Discussed:**

No.

**Experimental Designs Or Analyses:**

The experimental designs are well-structured, following prior works in continual learning. However, the hyperparameter sensitivity is not discussed well in this paper, such as the temperature coefficient.

**Methods And Evaluation Criteria:**

The proposed method avoids information leakage in continual learning and the benchmarks are appropriate for evaluation in continual learning.

**Other Comments Or Suggestions:**

There are several typos in the paper, for example, in lines 063-064, “L5-Large model” should be “T5-Large model”, and in line 192, “Equ 3” should be “Eq. 3” or “Eq(3)”.

**Other Strengths And Weaknesses:**

Strengths:
1. The proposed method avoids information leakage in continual learning, which is practical in real-world deployment.
2. The experiments are comprehensive to show the effectiveness of the proposed method, comparing against strong baselines, O-LoRA and SAPT.

Weaknesses:
1. The computational cost and memory overhead are not discussed. While the method avoids information leakage, it is unclear the cost for additional computation in similarity-based selection.
2. The empirical evidence is evident but this paper lacks rigorous theoretical analysis for the similarity metric selection, for example, the reason for choosing $L_2$ Euclidean distance and Dot Product Similarity rather than alternatives like cosine similarity.
3. The comparison discussion on previous works primarily focuses on SAPT and mentions the difference in information leakage. However, O-LoRA itself does not information leakage based on the definition of this paper since the key difference from O-LoRA is not clear and not discussed well.
4. The hyperparameter sensitivity is not discussed well in this paper. In the equation in line 217 on page 4, there is the temperature coefficient used to control normalization, but there is no analysis of the impact of this coefficient in the experiments.

**Questions For Authors:**

1. In the overall architecture of the proposed method, each task’s LoRA parameters and feature distributions are stored. Did authors compare the memory usage with O-LoRA? Since O-LoRA only needs to store the previous task LoRAs. Is there any rank reduction during the combination process in the right section in Figure 2? Could authors please compare the actual memory usage or training time with O-LoRA?
2. The authors in this paper define “information leakage”, which refers to the accessing or reusing of task-related information (e.g. training data and task identifiers) from previously learned tasks again. But in my opinion, the presentative feature distribution can still be considered indirect information, which is related to actual parameters. While in O-LoRA, it only uses the previous LoRA parameters. Could authors clarify the technique novelty compared to O-LoRA?
3. In Table 2, the AP result for O-LoRA on SuperNI benchmark is 37.17, but in SAPT paper, the AP result for O-LoRA on SuperNI benchmark is 24.26 (on page 20), which is a substantial difference. I checked the appendix of this paper and found that both studies use 50 training epochs and a learning rate of 5e-5. However, despite these similarities, the AP result in SAPT-LoRA with information leakage on SuperNI benchmark consistent similarities, whereas O-LoRA shows a significant difference in this paper. Could the authors clarify the reason for this discrepancy?
4. Since there is a temperature coefficient $T$ used to control normalization in the equation, could authors please provide the analysis of how different values of this coefficient impact the performance of the proposed method?

**Relation To Broader Scientific Literature:**

No specific.

**Theoretical Claims:**

This paper does not have any proof for theoretical claims, and the justification for using presentative feature distributions is mainly based on empirical experiments without solid theory formalization.

---

> ### Author Rebuttal · Authors · 2025-03-31
>
> We are grateful for the time and effort you have dedicated to reviewing our work. We provide point-by-point responses to address your concerns.
>
> **Q1: (Weaknesses) The empirical evidence is evident but ...... like cosine similarity.**
>
> A1: Due to word limit and the repeated mention of this weakness, we kindly ask you to refer to the responses for Q3 of Reviewer D8DQ and Q3 of Reviewer 54sH.
>
> **Q2: (Weaknesses) The hyperparameter sensitivity is not discussed ...... of this coefficient in the experiments.**
>
> A2: The temperature coefficient $T$ is an inherent hyperparameter of the softmax function. Following previous studies (SAPT, Zhao et al.), we fix this coefficient to 1.0. We add additional experiments concerning the impact of the temperature coefficient in the anonymous link https://anonymous.4open.science/r/ICML-2025-Rebuttal-10369/ICML_2025_Rebuttal.pdf (Table 17 and 18).
>
> In fact, the temperature coefficient and the $K$ have similar usage, as they both adjust the scaling of weights. When the $T$ methodes 0, it effectively corresponds to setting $K$ to 1. Therefore, it is not necessary to simultaneously adjust multiple parameters externally. We can only adjust one of them.
>
> **Q3: (Weaknesses and Questions) In the ...... compare the actual memory usage or training time with O-LoRA?**
>
> A3: We explicitly discussed the memory usage of our method in Lines 380-384 of the manuscript. To further clarify, we provide a detailed comparison across different methods in the anonymous link https://anonymous.4open.science/r/ICML-2025-Rebuttal-10369/ICML_2025_Rebuttal.pdf (Table 19).
>
> (Memory) While our method stores additional feature distributions compared with O-LoRA, these distributions are represented as lightweight vectors. Even when stored across all layers, they introduce only 0.262M additional parameters. As shown in Figure 3, retaining feature distributions for only a subset of layers achieves strong CL performance while further reducing memory overhead.
>
> (Training Time) SAPT and our method incur only a small increase in lightweight computations during training. However, the parameters added by SAPT require gradient updates and replay data, resulting in slower training speed for SAPT. In contrast, O-LoRA necessitates calculating the square difference between LoRA parameters during loss computation, which introduces substantial computational overhead and leads to the slowest training speed.
>
>
> There is no rank reduction during the combination process (illustrated in Figure 2). The feature distributions $D^{l} _{k} = \mathbb{E} _{p(x^{k}, y^{k})} [\mathbf{W}^{l}h^{l}(x^{k})]$ are derived from pre-trained model features and are independent of the LoRA blocks. This allows our method to seamlessly combine LoRA blocks of different ranks into a unified model.
>
> **Q4: (Questions) The authors define “information leakage” ...... Could authors clarify the technique novelty compared to O-LoRA?**
>
> A4: O-LoRA avoids catastrophic forgetting by ensuring that the parameters of new tasks are orthogonal to those of existing tasks. However, when new tasks overlap or conflict with learned tasks, this constraint can limit the learning performance. As a result, O-LoRA cannot perform as well as state-of-the-art continual learning methods, although it prevents IL as well as our method.
>
> Our method treats each LoRA block as a repository of knowledge, dynamically selecting them as needed. Although our method preserves the presentative feature distributions, these distributions are high-dimensional and beyond human comprehension, similar to the LoRA parameters. Moreover, presentative feature distributions represent averaged information across populations and cannot reconstruct individual information. Therefore, our approach achieves superior continual learning performance while effectively avoiding information leakage.
>
> **Q5: (Questions) In Table 2, the AP result ...... Could the authors clarify the reason for this discrepancy?**
>
> A5: To ensure a fair comparison, we implemented previous CL methods using the open-source framework of SAPT. These adjustments led to improved performance for most methods compared with their original reports. Our experiments are based on the SAPT framework, which explains the lack of significant differences in SAPT results as reported. It's important to note that SAPT did not provide detailed instructions for reproducing O-LoRA. Our settings may differ due to two key hyperparameters in O-LoRA: we consistently set $\lambda_{1}$ to 0.5 and $\lambda_{2}$ to 0, retaining all LoRA layers without merging them—factors that significantly impact results. Additionally, discussions on the O-LoRA GitHub indicate that the number of GPUs used can influence results, with an error margin exceeding 8\%. Our experiments utilized 8 A100 GPUs, while SAPT used 4 A800 GPUs. We provide the code to reproduce the O-LoRA results in the anonymous link https://anonymous.4open.science/r/ICML-2025-Rebuttal-10369/ICML_2025_Rebuttal.pdf.

---

> > ### Comment · Reviewer_MixK · 2025-04-02
> >
> > Thank you for providing additional experiments and explanations in response to my questions and concerns. I have one follow-up question:
> >
> > Based on Table 15 and Table 16 in the rebuttal, there does not appear to be a significant difference in accuracy performance between cosine similarity and $L_2$ similarity. Given this observation, could cosine similarity also be considered an alternative for your proposed method?

---

> > > ### Author Response · Authors · 2025-04-03
> > >
> > > Thank you so much for your prompt reply!
> > >
> > > We agree with you that there does not exist a significant difference in accuracy performance between the cosine similarity and the $L_2$ similarity. Hence the cosine similarity can be considered an alternative for the $L_2$ similarity used in our method, which means, we can optionally use the cosine similarity in our method. **Actually, this also means that our proposed method does not reply on a specific similarity measure, which makes our method more flexible with plug-in similarity measures.** This advantage allows our method to choose a suitable similarity measure based on the specific performance, enabling it to be effectively applied across a broader range of scenarios.
> > >
> > > It is noteworthy that we have shown the performance of the dot similarity, the $L_2$ similarity, and the cosine similarity. In fact, any metric that evaluates the similarity degree between two vectors can be used as a plug-in similarity measure for our method. **Terefore, we consider that this is also an important advantage of our method for practical use, in addition to avoiding information leakage.**
> > >
> > > Thank you again for your insightful comments and we are very willing to discuss with you if you have any other concerns.

---

### Official Review · Reviewer_D8DQ · 2025-03-14

**Overall Recommendation:** 2

**Summary:**

This paper presents a method for continual learning (CL) in large language models (LLMs) that addresses information leakage (IL) while maintaining strong performance. The method leverages the feature representation capability of pre-trained LLMs to encode task-related information into presentative feature distributions, and dynamically selects relevant LoRA blocks based on the similarity between input instances and the presentative feature distributions of different tasks. Experiments demonstrate the effectiveness of the method across multiple benchmarks and model architectures.

**Claims And Evidence:**

The claims are clear and supported by convincing evidence.

**Essential References Not Discussed:**

CL methods with similar ideas to this paper but not discussed include iCaRL (Rebuffi et al., 2017), Feature Adaptation (Iscen et al., 2020), etc. There are also parameter isolation-based LLM methods that haven’t been discussed in this paper, e.g. MoLoRA (Zadouri et al., 2023) and LoRAMoE (Dou et al., 2023).

**Experimental Designs Or Analyses:**

The experimental settings are reasonable. The choice of benchmarks, baseline methods and evaluation metrics are appropriate and comprehensive. However, the effectiveness of different modules (i.e. presentative feature distribution and dynamic LoRA selection) have not be sufficiently demonstrated by e.g. ablation studies. In addition, in Table 3, SAPT-LoRA without IL performs even better than the one with IL, which is contrary to the previous results. It would be better if the authors attempt to explain this phenomenon in the paper.

**Methods And Evaluation Criteria:**

The paper proposes a CL method for LLMs that addresses the IL issue by avoiding introducing new trainable parameters during the selection process. Although the method does prevent catastrophic forgetting and IL, it lacks novelty. Similar methods have already been proposed by earlier works. For example, iCaRL (Rebuffi et al., 2017) uses a nearest-mean-of-examplars classifier, which is highly similar with the selection module in this paper. The idea of storing feature distributions instead of training data has also appeared in previous works, e.g. Feature Adaptation (Iscen et al., 2020).

**Other Comments Or Suggestions:**

There are some typos in the paper:
Table 1 – “L5” should be “T5”
Table 4 – “Singe” should be “Single”
Table 4 – “proformance” should be “performance”

**Other Strengths And Weaknesses:**

The authors mentioned that IL hinders the application of CL in scenarios involving data-sensitive or specialized tasks. It would be better if they could provide some examples – what kind of scenarios in practice would requires high data sensitivity or privacy, why such scenarios require CL, and how CL can be applied to these scenarios?

**Questions For Authors:**

1. Application in practice: what kind of scenarios in practice would IL matter, why such scenarios require CL, and how CL can be applied to these scenarios?
2. Does your method support sparse activation like MoE (i.e. only activate some of the LoRA modules in a layer, instead of all at once)? The formula in Section 3.4 suggests that it’s not supported (since all LoRAs contribute to the final output).
3. What’s the difference between cosine similarity and the dot product similarity you propose in Section 3.3?
4. In the zero-shot experiment (Table 3), why would SAPT-LoRA without IL perform better than the one with IL?

**Relation To Broader Scientific Literature:**

This paper advances the field by addressing a specific limitation (information leakage) in CL for LLMs. It contributes a solution to a practical problem – how to apply CL methods of LLMs to data-sensitive scenarios. It also presents a conceptual framework for how pre-trained models can be adapted in ways that respect data privacy constraints.

**Theoretical Claims:**

This paper presents no proofs or theoretical claims.

---

> ### Author Rebuttal · Authors · 2025-03-31
>
> Thank you so much for your valuable comments! We provide point-by-point responses to address your concerns.
>
> **Q1: (Methods and References) Similar methods have already been proposed ...... e.g. Feature Adaptation (Iscen et al., 2020).**
>
> A1: Thank you for pointing out these relevant studies, and we will incorporate them into our related work. They indeed utilize feature representations for incremental learning. **However, our method is significantly different from theirs.**
>
> Firstly, we focus on LLMs and aim to make them to continually learn (fine-tune) in dynamic environments, such as distribution shifts or integrating new knowledge. In contrast, iCaRL and Feature Adaptation primarily address classification tasks, where the goal is to increase the number of classes that the model can recognize, which is fundamentally different from our goal.
>
> Secondly, iCaRL utilizes the distance between the output features and the centers of each class. In our selection module, we utilize distances between **pre-trained features** and task center from **each layer**. The sources of features and the targets of selection are different.
>
> Additionally, Feature Adaptation preserves features extracted by different feature extractors and embeds them to the same space. In contrast, we directly utilize the feature from frozen pre-trained LLMs. Moreover, we do not claim to be the first to use feature distributions as a surrogate for training data. Our goal is to employ these distributions to avoid **information leakage**.
>
> **Q2: (Experiments) The effectiveness of different modules have not be sufficiently demonstrated by e.g. ablation studies.**
>
> A2: Our method first leverages pre-trained LLMs to encode tasks into presentative feature distributions, and then calculate the similarity between instances and the presentative feature distributions to dynamically select proper PEFT blocks. In other words, presentative feature distribution and dynamic selection are connected modules of our method that must work in sequence. Removing any module from our method would make it meaningless.
>
> **Q3: (Weaknesses and Questions) The authors mentioned that IL ......  and how CL can be applied to these scenarios?**
>
> A3: In most cases, model weights (instead of training data) are commonly shared (e.g., open-source ecosystem DeepSeek, Qwen and Mistral). In such situations, CL through data replay is not feasible. Enterprises and research institutions often fine-tune models for specialized tasks with training data that contains proprietary information, such as medical assistants and fraud detection systems. This data often includes user interactions, which cannot be shared due to privacy regulations. This presents challenges for CL, such as adapting models to evolving regulations or updating fraud detection systems to counter new threats, especially when these models must integrate prior knowledge. In these scenarios, effective CL without IL is crucial.
>
> **Q4: (Questions) Does your method support sparse activation like MoE ...... (since all LoRAs contribute to the final output).**
>
> A4: Of course, as our method is based on parameter isolation techniques, it inherently supports selective activation similar to MoE. In fact, we have already implemented this method. As demonstrated in Figure 4, we present the performance variation curve using Top-$k$ activation, which only activates the $k$ closest LoRA modules. We recommend selecting a smaller $k$ to achieve more stable results.
>
> **Q5: (Questions) What’s the difference between cosine similarity ...... you propose in Section 3.3?**
>
> A5: Cosine similarity $\text{Cos}(A, B) = \frac{A \cdot B}{||A|| ||B||}$ and dot product similarity $\text{Dot}(A, B) = \frac{A \cdot B}{\sqrt{d}}$ are both mathematical methods to measure the similarity degree between two vectors. Their primary difference lies in the normalization scale ($||A|| ||B||$ for cosine similarity and $\sqrt{d}$ for dot product similarity).
> We also conducted experiments using cosine similarity. Since dot product similarity is commonly used in the attention calculation, we reported its results in our manuscript. You can find the experiments of cosine similarity through the anonymous link https://anonymous.4open.science/r/ICML-2025-Rebuttal-10369/ICML_2025_Rebuttal.pdf (Table 15 and 16).
>
> **Q6: (Questions) In the zero-shot ...... why would SAPT-LoRA without IL perform better than the one with IL?**
>
> A6: We have indeed observed this phenomenon. It might be due to data replay causing the model to overfit, which results in reduced generalization capabilities. Since the replay data in SAPT is synthetic, it primarily emphasizes the prompt and structure of the problem. As a result, in a zero-shot setting, it is likely to overlook the knowledge required to select the LoRA blocks.

---

### Decision · Program_Chairs · 2025-05-01

**Decision:**

Accept (poster)

**Comment:**

This paper solves an important problem in LLM continual learning, called information leakage, where task-related information of learned tasks is accessed or reused again. This problem could impose potential risks on data privacy, which might hinder the real-world deployment of LLMs. To address this problem, this paper proposes to exploit presentative feature distributions by pre-trained LLMs and use them to dynamically select relevant knowledge to prevent forgetting. This method is simple and intuitive, but it is effective and has multiple advantages, for example, avoiding information leakage, no trainable parameters, and flexible expansion. Experimental results can demonstrate the effectiveness of the proposed method.

During the review process, this paper received mixed reviews, containing 1 accept and 2 weak reject, from 3 reviewers. All the reviewers confirmed the proposed method can avoid information leakage in LLM continual learning, and experimental results can valid the effectiveness of the method. However, Reviewer D8DQ has major concerns in the novelty and the application of the method. Reviewer MixK also raised major concerns in the similarity and training time. Reviewer 54sH gave positive rating and asked some minor questions.

During the rebuttal process, the authors provided a rebuttal, and all the reviewers acknowledged it. Reviewer 54sH said no further questions and decided to keep acceptance. Reviewer MixK raised one follow-up question about whether cosine similarity could also be considered an alternative for the proposed method. The authors replied to this question, while Reviewer MixK didn’t reply whether his follow-up question is addressed or not. Reviewer D8DQ also didn’t say whether his concerns are addressed or not.

From my viewpoint, this paper does not reply on certain evaluation metrics, so cosine similarity can be also used, and this is an advantage instead of disadvantage. I feel although the method is simple, the authors’ rebuttal explained the differences between this work and previous work. Since this work can avoid information leakage, the application would be more promising than similar methods. So I think the authors’ rebuttal addressed the concerns of Reviewer MixK and Reviewer D8DQ. So I recommend acceptance.